# Five energy metabolism pathways show distinct regional distributions and lifespan trajectories in the human brain

**Moohebat Pourmajidian**[1], **Justine Y. Hansen**[1], **Golia Shafiei**[2],
**Bratislav Misic**[1], **Alain Dagher**[1]*

1 Montréal Neurological Institute, McGill University, Montréal, Quebec, Canada, 2 Department of Psychiatry, Perelman School of Medicine, University of Pennsylvania, Philadelphia, Pennsylvania, United States of America

☙ These authors contributed equally to this work.
* alain.dagher@mcgill.ca.

## Abstract

Energy metabolism involves a series of biochemical reactions that generate ATP, utilizing substrates such as glucose and oxygen supplied via cerebral blood flow. Energy substrates are metabolized in multiple interrelated pathways that are cell- and organelle-specific. These pathways not only generate energy but are also fundamental to the production of essential biomolecules required for neuronal function and survival. How these complex biochemical processes are spatially distributed across the cortex is integral to understanding the structure and function of the brain. Here, using curated gene sets and whole-brain transcriptomics, we generate maps of five fundamental energy metabolic pathways: glycolysis, pentose phosphate pathway, tricarboxylic acid cycle, oxidative phosphorylation and lactate metabolism. We find consistent divergence between primarily energy-producing and anabolic pathways, particularly in unimodal sensory cortices. We then explore the spatial alignment of these maps with multi-scale structural and functional attributes, including metabolic uptake, neurophysiological oscillations, cell type composition, laminar organization and macro-scale connectivity. We find that energy pathways exhibit unique relationships with the cellular and laminar organization of the cortex, pointing to the higher energy demands of large pyramidal cells and efferent projections. Finally, we show that metabolic pathways exhibit distinct developmental trajectories from the fetal stage to adulthood. The primary energy-producing pathways peak in childhood, while the anabolic pentose phosphate pathway shows greater prenatal expression and declines throughout life. Together, these results highlight the rich biochemical complexity of energy metabolism organization in the brain.

**Data availability statement:** All the scripts and data used to perform the analyses are available at https://github.com/netneurolab/pourmajidian_metabolism-genes/ and on Zenodo at https://zenodo.org/records/17983812 (DOI: https://doi.org/10.5281/zenodo.17983812. Biological databases are openly accessible in *biomaRt* at https://bioconductor.org/packages/release/bioc/html/biomaRt.html. The Allen Human Brain Atlas is available at https://human.brain-map.org/static/download/. The source data from the Human Connectome Project S900 release including diffusion-weighted MRI, functional MRI and MEG are available at https://db.humanconnectome.org/. Group-averaged PET, MEG and functional connectivity gradient images are available in *neuromaps* at https://netneurolab.github.io/neuromaps/. The mitochondrial profile maps are available at https://neurovault.org/collections/16418/. BrainSpan RNA-sequencing and microarray datasets are available at https://www.brainspan.org/static/download.html. The Schaefer-400 and 100 parcellations are openly available at https://github.com/ThomasYeoLab/CBIG/tree/master/stable_projects/brain_parcellation/Schaefer2018_LocalGlobal/Parcellations/ and can also be retrieved using *abagen* at https://abagen.readthedocs.io/. The Glasser parcellation is available at https://github.com/brainspaces/glasser360.

**Funding:** AD acknowledges support from the Canadian Institutes of Health Research (CIHR) Foundation scheme (FDN-143242, https://cihr-irsc.gc.ca/). BM acknowledges support from the Natural Sciences and Engineering Research Council of Canada (NSERC, RGPIN-2017-04265, https://nserc-crsng.canada.ca), Canadian Institutes of Health Research (CIHR, PJT-180439), Brain Canada Foundation

## Introduction

The brain relies on substantial energy to maintain signaling and housekeeping functions. Generation of action potentials, synaptic activity, neurotransmitter release, uptake and repackaging rely on energy production via a multitude of chemical pathways [1–3]. These fundamental and interrelated biological pathways transform nutrients to generate adenosine triphosphate (ATP), the main energy currency within the cell.

Energy metabolism is a dynamic process that changes from early development to adulthood and aging, reflecting shifts in substrate utilization and metabolic regulation to meet evolving cellular energy demands [4]. The main source of energy in the adult human brain is glucose [5,6]. However, the brain can also utilize alternative energy sources including lactate, ketone bodies and fatty acids under certain physiological circumstances such as intense physical activity, fasting and at specific developmental stages [4,7,8]. Uniquely among all organs, the brain stores virtually no energy, with only minimal glycogen reserves located predominantly in astrocytes [9,10]. Brain cells therefore rely on a constant nutrient supply from the vasculature and energy production coupled to synaptic activity. As a result, energy metabolic pathways are tightly regulated and dynamically adapt to changes in nutrient supply and demand.

Once glucose is taken up by the brain, it can be metabolized via multiple interacting metabolic pathways (Fig 1). Glycolysis is the first step in the breakdown of glucose. It converts one molecule of glucose to two molecules of pyruvate, while producing a net of two ATP molecules. Lactate dehydrogenases can catalyze the interconversion of pyruvate and lactate, regulating the cellular redox state (maintaining the $NAD^+$:NADH balance), and provide lactate as an energy substrate that can be exported, taken up from the extracellular space, or converted to pyruvate to enter the downstream energy pathways [11]. Alternatively, glucose can enter the pentose phosphate pathway (PPP), an anabolic pathway essential for cellular biosynthesis. The PPP produces 5-carbon sugars for subsequent nucleotide, amino acid and neurotransmitter synthesis and generates nicotinamideadenine-dinucleotide phosphate (NADPH), an essential cofactor for lipid biosynthesis and cellular defense against reactive oxygen species (via glutathione synthesis) [7,9,12]. Furthermore, pentose sugars produced via PPP can be metabolized into glycolytic intermediates and subsequently converted to pyruvate by glycolytic enzymes [13,14].

Pyruvate can be shuttled to the mitochondrial matrix where it ultimately enters the tricarboxylic acid cycle (TCA) to produce high-energy electron carriers nicotinamide adenine dinucleotide (NADH) and Flavin adenine dinucleotide (FADH2). The TCA cycle is also involved in anabolic processes by providing precursors for amino acid, nucleotide and fatty acid synthesis and neurotransmitters such as glutamate and gamma-aminobutyric acid (GABA) [2,4,15]. The high-energy electron carriers NADH and FADH2 can then enter the electron transport chain (ETC) within the mitochondrial inner membrane. The ETC is made up of four protein complexes that transfer electrons from NADH and FADH2 to molecular oxygen, producing a proton gradient across the mitochondrial membrane. ATP synthase, the fifth mitochondrial complex,

Future Leaders Fund, the Canada
Research Chairs Program
(CRC-2022-00169, https://www.chairs-
chaires.gc.ca/), the Michael J. Fox
Foundation (MJFF-021133,
https://www.michaeljfox.org/), and the
Healthy Brains, Healthy Lives initiative
(HBHL, https://www.mcgill.ca/hbhl/). MP
acknowledges support from the HBHL
and the Harold and Audrey Fisher
Training Studentship (https://www.mcgill.
ca/neuro/training/financial-
support/studentships). JYH acknowledges
support from the Helmholtz International
BigBrain Analytics and Learning
Laboratory (https://bigbrainproject.
org/hiball.html), the NSERC and the
Neuro-Irv and Helga Cooper Foundation
(https://www.mcgill.ca/neuro/open-
science/open-science-awards-and-
prizes/neuro-irv-and-helga-cooper-
foundation-open-science-prizes). GS was
supported by a postdoctoral fellowship
from the CIHR.The funders had no role in
study design, data collection and analysis,
decision to publish or preparation of the
manuscript.

**Competing interests:** The authors have
declared that no competing interests exist.

uses this electrochemical gradient to produce ATP, completing the oxidative phosphorylation pathway (OXPHOS). OXPHOS enables the complete oxidation of glucose, producing ~15 times more ATP than glycolysis per molecule of glucose. Due to its high efficiency in ATP production, OXPHOS is regarded as the primary pathway for ATP generation in the brain [5]. Interestingly, mitochondria are also a major source of reactive oxygen species (ROS) due to electron leakage from the ETC [16, 17]. Collectively, glucose metabolic pathways are not only essential for ATP production, but also play pivotal roles in anabolic processes and antioxidant homeostasis and are fundamental to cellular growth, repair, and survival.

A growing body of evidence suggests a compartmentalization of energy metabolism across the different cell types of the brain [18–20]. The differential expression of genes and enzymes, along with the selective distribution of transporters involved in energy metabolism, suggests that astrocytes favor glycolysis, whereas neurons rely more on oxidative metabolism [10,11,21–23]. However, this metabolic division remains a topic of ongoing debate [6,18] and the energy profiles and critical contributions of other glial cell types such as oligodendrocytes and microglia remain relatively understudied and are only beginning to be understood [24–26].

Multiple imaging modalities have been employed to map energy metabolism in the human brain. Positron Emission Tomography (PET) has been instrumental in studying glucose uptake and oxygen consumption [27–31]. However, PET radiotracers do not provide the biochemical resolution required to distinguish between downstream glucose metabolic pathways. Magnetic resonance spectroscopy (MRS) has also been used to study energy metabolism [32–36]. MRS enables the measurement of metabolic pathway rates and metabolite concentrations; however, its low sensitivity and limited spatial resolution do not allow for precise characterization and mapping of metabolic pathways in the whole brain [37,38].

To map energy metabolism in the brain at a resolution that would allow pathway-specific insights, we employ neuroimaging transcriptomic techniques in postmortem human brains [39]. This approach provides a link between molecular data and the structural and functional architecture, allowing for a unified framework to study energy metabolism in the brain. Here, we map the distinct gene expression profile of five fundamental energy metabolism pathways across the cortex. We further explore their spatial correspondence to multi-scale structural and functional cortical features, and chart their developmental trajectories through the human lifespan.

## Results

We use whole-brain microarray gene expression from the Allen Human Brain Atlas (AHBA) [41] to generate maps of five key energy metabolism pathways including: glycolysis, pentose phosphate pathway (PPP), tricarboxylic acid cycle (TCA), oxidative phosphorylation (OXPHOS), and lactate metabolism and transport. Briefly, gene sets for each pathway were identified based on their corresponding Gene Ontology (GO) biological process [42] and Reactome pathway [43] IDs and further filtered to only retain pathway-annotated genes included in both databases. Pathway IDs and final

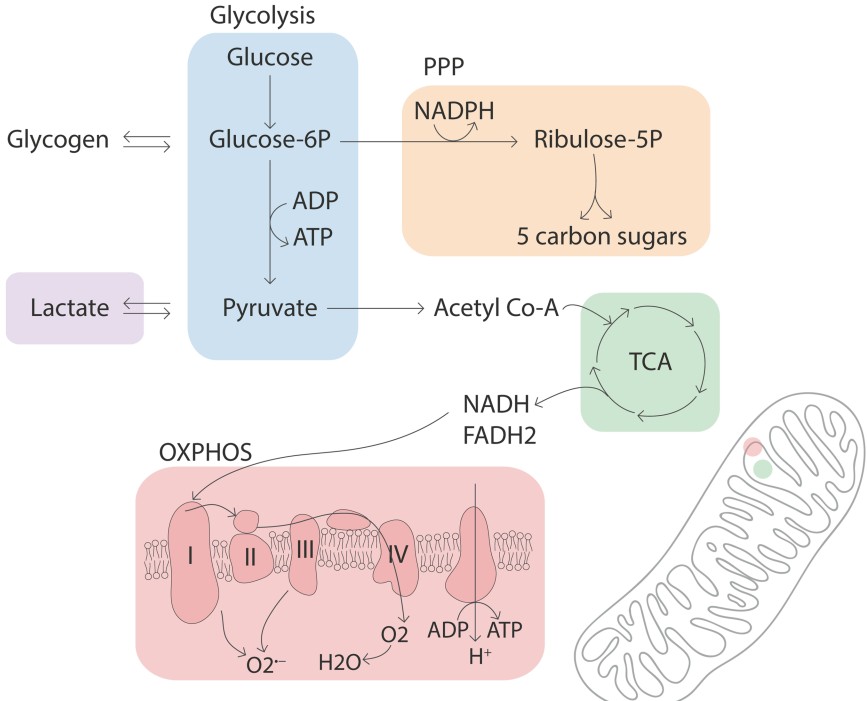

**Fig 1**. **Pathways involved in brain energy metabolism.** Energy metabolism refers to processes involved in energy production from nutrient molecules. Glucose is the main energy source in the brain under normal physiological conditions. Glucose entering brain cells can be utilized in three parallel pathways. It can be processed through glycolysis to produce 2 ATP and 2 pyruvate molecules (blue). Lactate dehydrogenases catalyze the inter-conversion of lactate and pyruvate (purple). Pyruvate is transported into mitochondria, where it enters the TCA cycle to generate high-energy electron carriers NADH and FADH2 (green), driving the complete oxidation of glucose through the mitochondrial electron transport chain and oxidative phospho-rylation (red). Glucose entering the brain can also enter the PPP (orange). PPP is an anabolic pathway that uses glucose to produce 5-carbon sugars and NADPH, an essential co-factor used in nucleotide and lipid biosynthesis (yellow). Glucose can also be stored in the form of glycogen via glycogen synthase, a process mainly active in astrocytes. PPP, pentose phosphate pathway; TCA, tricarboxylic acid cycle; OXPHOS, oxidative phosphorylation; NADH, nicotinamide adenine dinucleotide; FADH2, Flavin adenine dinucleotide.

gene sets used to produce the energy maps are provided in S1 Table. Microarray transcriptome data were retrieved using the *abagen* package and parcellated into the Schaefer-400 cortical atlas ([40,44], https://abagen.readthedocs.io/). Expression was then averaged across all genes to produce a mean gene expression map for each energy pathway.

## Mapping metabolic pathways using gene expression

Fig 2A shows a Venn diagram of the final number of genes included in each energy pathway expression matrix and the number of shared genes between them. Note that the final gene sets for each pathway contain fewer genes compared to the original gene sets retrieved from the GO and Reactome databases, as some of the genes are not present in the AHBA or did not meet quality control criteria (see *Methods*). It should be noted that the OXPHOS gene set is predominantly pop-ulated by mitochondrial complex I genes given its substantially larger size (45 subunits) compared to other complexes [45]. Additionally, mitochondrially-encoded genes were not included in the AHBA microarray platform and are therefore absent from our pathway maps (S7 Table).

Importantly, there is minimal overlap between the different energy pathway gene sets, which allows the reconstruction of distinct maps that represent each pathway individually, facilitating their study in relative isolation despite their inherent interconnectivity. To assess the correspondence between metabolic pathway maps, we first compute spatial correlations

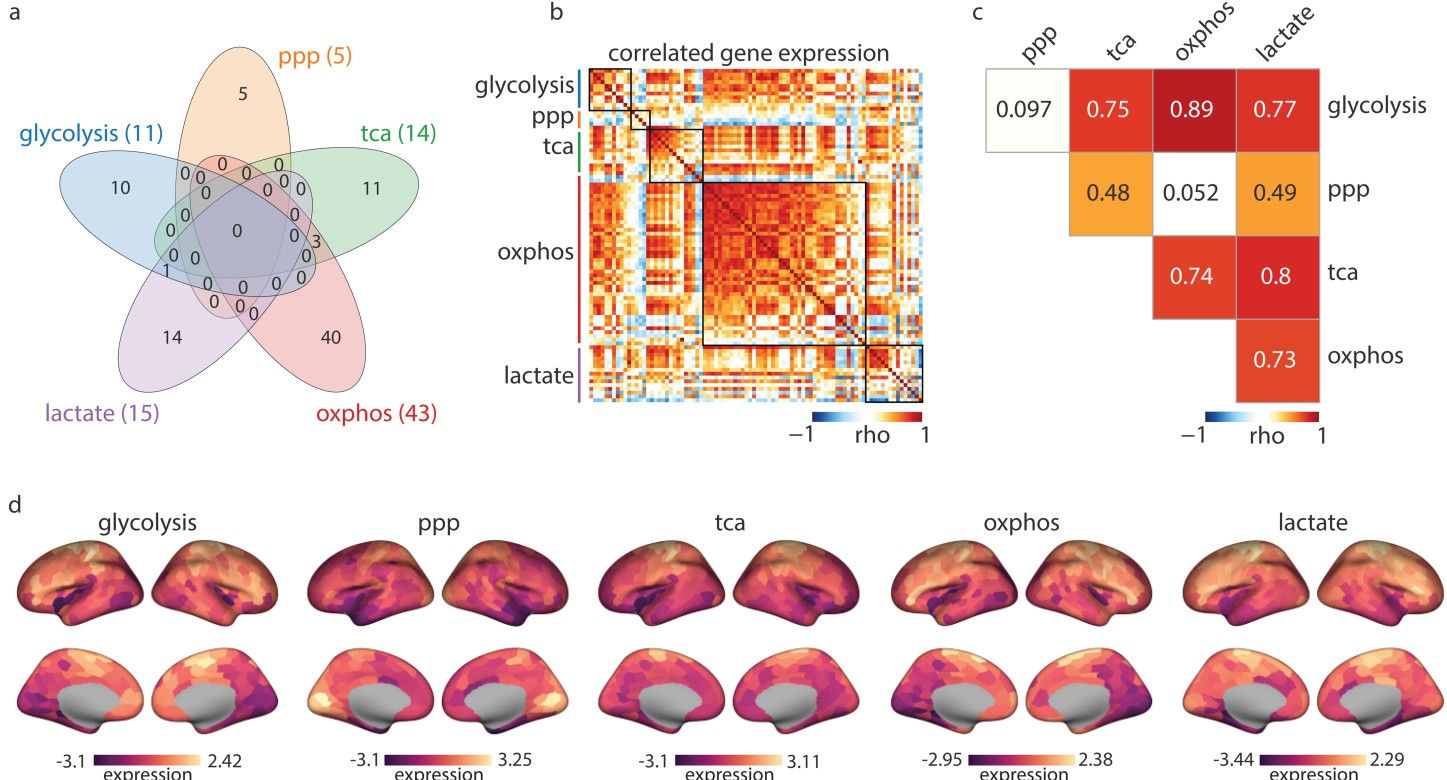

**Fig 2. Brain maps of energy metabolism pathways.** (a) Left: the Venn diagram shows the final number of genes in each pathway gene expression matrix and the number of shared genes between pathways. Gene sets have minimal overlap; there are three genes shared between the TCA and OXPHOS belonging to the succinate dehydrogenase complex that is active in both pathways. *PFKFB2*, which encodes a regulatory enzyme of the glycolytic pathway, is shared between the glycolysis and lactate pathways. The full name of the genes in each pathway can be found in S1 Table. (b) Heatmap depicts the pairwise correlation of all genes in the energy pathways across the 400 cortical regions in the Schaefer parcellation [40]. (c) Spearman's correlation among mean expression energy maps. (d) Energy pathway maps. Colorbar shows z-scored mean expression values across all genes in each pathway. The data underlying this figure can be found in S1 Data. For the subcortical energy pathway profiles see S3 Fig. ppp, pentose phosphate pathway; tca, tricarboxylic acid cycle; oxphos, oxidative phosphorylation; lactate, lactate metabolism and transport.

among them. Fig 2B shows the correlated gene expression for genes in all energy pathways across 400 cortical regions, and Fig 2C shows global spatial correlations among mean expression energy maps.

Glycolysis and OXPHOS maps show the strongest correlation (rho = 0.89, $p_{spin} = 1 \times 10^{-4}$) consistent with the fact that they are part of an integrated sequence of events in the oxidation of glucose [46,47]. The lowest correlations are observed between the PPP map and the glycolysis and OXPHOS maps (rho = 0.097, $p_{spin} = 0.79$; rho = 0.052, $p_{spin} = 0.91$, respectively), in line with the role of PPP as a primarily biosynthetic pathway rather than one directly implicated in energy production [14]. Conversely, given the PPP's function in cellular defense against ROS, it could be anticipated that regions with high oxidative metabolism would exhibit relatively elevated PPP activity. However, after quality control, the PPP gene set predominantly includes genes involved in its anabolic non-oxidative branch (i.e. *RPEL1*, *RPIA*, *PRPS2*, *RBKS*), responsible for the production of 5-carbon sugars and downstream nucleotide synthesis [6,48]. Indeed, when we look into the *PGD* gene, the only gene representing the oxidative branch in our PPP gene set, which also directly catalyzes one of the the NADPH-producing reactions, we find significant spatial correlations with glycolysis and OXPHOS maps (glycolysis: rho = 0.75, $p_{spin} = 1 \times 10^{-4}$; OXPHOS: rho = 0.71, $p_{spin} = 1 \times 10^{-4}$; S1 Fig).

The PPP map shows a moderate correlation with the TCA map (rho = 0.48, $p_{spin}$ = 0.05), potentially reflecting their shared roles in supporting cellular anabolic processes, and highlighting cortical regions with greater biosynthesis demands. Furthermore, the TCA map also shows a strong alignment with the glycolysis and OXPHOS maps (rho = 0.75, $p_{spin}$ = $1 \times 10^{-4}$; rho = 0.74, $p_{spin}$ = $1 \times 10^{-4}$, respectively). These associations highlight the dual role of the TCA in energy production and cellular biosynthesis [15,20,49]. The lactate map shows strong correlations with glycolysis (rho = 0.77, $p_{spin}$ = $1 \times 10^{-4}$), TCA (rho = 0.8, $p_{spin}$ = $2 \times 10^{-4}$), and OXPHOS (rho = 0.73, $p_{spin}$ = $1 \times 10^{-4}$), likely reflecting lactate's role as a versatile intermediate in brain energy metabolism. Lactate, produced either via aerobic glycolysis or taken up from the vasculature, is converted to pyruvate by lactate dehydrogenase and readily utilized in the TCA cycle [11,46]. The spatial alignment observed between lactate and the other energy pathway maps could underscore its role in shuttling metabolic substrates across pathways, linking glycolysis and oxidative metabolism. Collectively, these results highlight the dependencies and interplay between energy pathways. In the following section we investigate the regional heterogeneity of these pathways and their enrichment across structural and functional systems.

## Regional heterogeneity of metabolic pathways

How are metabolic pathways distributed across the cortex? Fig 2D shows that energy pathway gene expression is regionally heterogeneous. Glycolysis and OXPHOS exhibit greater expression in motor and prefrontal cortex, and lower expression in the visual cortex. In contrast, the PPP map shows greater expression in the visual cortex. The TCA map shows a particularly higher expression in somato-motor regions (Figs 2D and S2). Across the subcortical regions, energy maps consistently show greater expression in the thalamus and lower expression in the amygdala [55] (S3 Fig).

To investigate how the spatial patterning of metabolic pathways aligns with the structural and functional organization of the brain, we estimate their enrichment in four atlases: (1) cytoarchitecture (von Economo-Koskinas classes; [50,52]), (2) synaptic and laminar architecture (Mesulam classes; [53,56,57]), (3) unimodal-transmodal hierarchy [54,56], and (4) intrinsic functional networks (Yeo-Krienen networks; [40,58]). For each network class, the average expression of parcels falling into that class was calculated and tested against a distribution of 10 000 spatial autocorrelation-preserving nulls. Across the seven von Economo-Koskinas cytoarchitectonic classes, all energy pathway maps except for the PPP have significantly higher expression in the primary motor cortices (glycolysis: $p_{spin}$ = 0.01; TCA: $p_{spin}$ = $8 \times 10^{-4}$; OXPHOS: $p_{spin}$ = 0.008; lactate: $p_{spin}$ = 0.002; Fig 3). The PPP map shows significantly greater average expression in the primary sensory cortex ($p_{spin}$ = $1 \times 10^{-4}$). The insular cortex has the lowest expression across all energy maps with PPP, TCA and lactate maps showing significantly lower values (PPP: $p_{spin}$ = 0.003; TCA: $p_{spin}$ = $4 \times 10^{-4}$; lactate: $p_{spin}$ = 0.04).

To investigate how the energy maps align with the broader cortical hierarchy and functional organization, we further estimated their enrichment across the Mesulam sensory-fugal hierarchy [56] and the first two principal gradients of resting state functional connectivity (FC) [54]. Along the Mesulam sensory-fugal axis, the PPP, TCA and lactate maps exhibit a similar enrichment, with significantly greater expression in idiotypic areas (PPP: $p_{spin}$ = 0.002; TCA: $p_{spin}$ = 0.002; lactate: $p_{spin}$ = 0.001), and overall lower expression in the paralimbic regions. Glycolysis and OXPHOS maps do not show significant variations across this hierarchy. Likewise, the first FC gradient (FC1) significantly correlates with the PPP and TCA maps (PPP: rho = 0.47, $p_{spin}$ = $5 \times 10^{-4}$; TCA: rho = 0.37, $p_{spin}$ = 0.03) corresponding to overall greater expression in primary cortices and lower expression in higher order association areas. The second FC gradient (FC2), which differentiates within the primary cortices, correlates significantly with the glycolysis (rho = 0.47, $p_{spin}$ = 0.016) and OXPHOS maps (rho = 0.46, $p_{spin}$ = 0.01), reflecting greater expression in motor regions and lower expression in the occipital cortex. S4 Fig shows similar findings using the finer delineation of these functional patterns according to the Yeo-Krienen intrinsic networks. Collectively, these results highlight the heterogeneous distribution of energy metabolism pathways across the structural and functional organization of the cortex, and point to a consistent dichotomy between pathways primarily involved in ATP production (glycolysis and OXPHOS) and the anabolic PPP. The TCA cycle, which contributes to both oxidative metabolism and anabolic processes, integrates features of both metabolic functions.

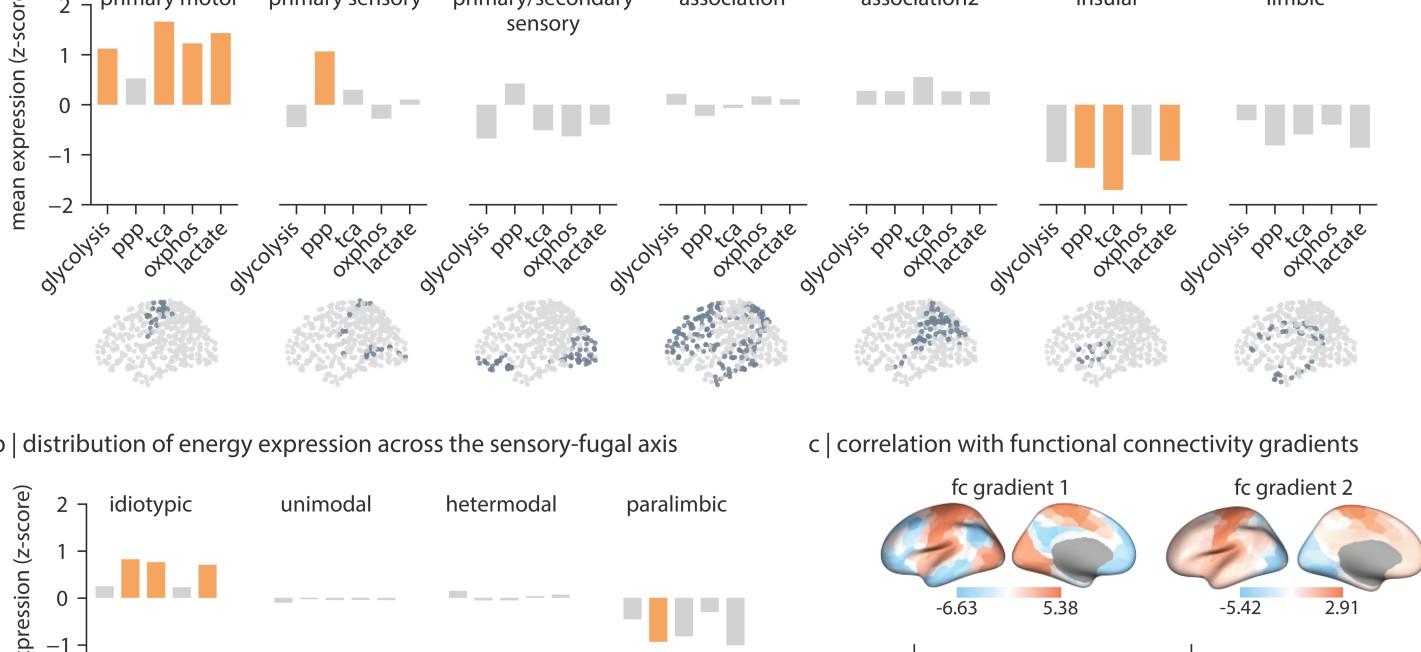

a | distribution of energy expression across cytoarchitectonics classes

b | distribution of energy expression across the sensory-fugal axis

c | correlation with functional connectivity gradients

■ $p_{spin} < 0.05$

**Fig 3. Distribution of energy pathway maps across structural and functional networks in the cortex.** For each energy map, average expression values for parcels falling into each structural class and functional network was calculated. (a) Distribution of energy pathway mean expression across seven von Economo-koskinas cytoarchitectonic classes [50–52]. (b) Distribution of energy maps across the sensory-fugal axis of information processing [53]. Bars represents observed average expression of each energy map in each network class. The y-axis represents mean gene expression of z-scored maps. (c) Spearman's correlation between energy maps and the first two functional connectivity (FC) gradients in the human cortex [54]. Highlighted bars indicate statistical significance when tested against 10 000 spatial-autocorrelation preserving nulls ($p_{spin} < 0.05$). Brain plots visualize regions included in each network class. The data in this figure can be found in S1 Data. ppp, pentose phosphate pathway; tca, tricarboxylic acid cycle; oxphos, oxidative phosphorylation; lactate, lactate metabolism and transport; fc, functional connectivity.

## Energy gradients in the visual cortex

The visual cortex stands out in our analyses as one of the regions in which energy pathways are most differentiated, with greater expression of PPP and lower expression of glycolysis and OXPHOS. However, previous research characterizes the visual cortex as having higher glucose and oxygen consumption [60,61]. This discrepancy may be due to the underlying heterogeneity within the visual cortex, where individual sub-regions exhibit distinct metabolic profiles that are not captured when analyzed as a single structure. Here, we investigate the distribution of the energy maps across the information processing hierarchy of the visual cortex as defined by the Glasser atlas [59] (S2 Table). TCA, OXPHOS and lactate show greater expression in the dorsal stream and lower expression in the ventral stream (Fig 4). Glycolysis also exhibits greater

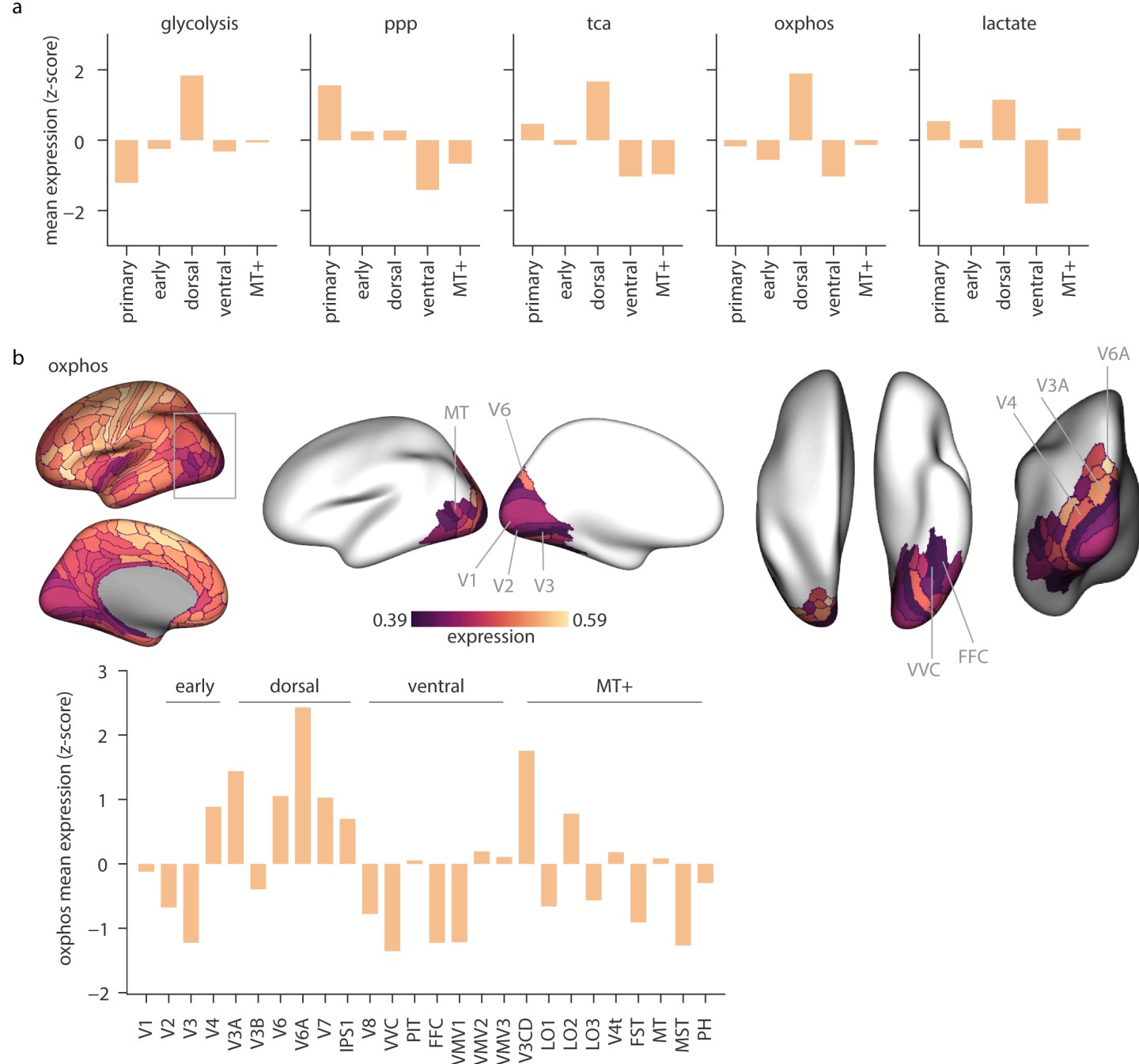

**Fig 4. Spatial distribution of energy pathways within the visual cortex.** Energy maps were produced using the Glasser parcellation and the visual parcels were grouped according to the visual information processing hierarchy detailed in [59] (see S2 Table). (a) Distribution of energy pathway mean expression across the visual hierarchy. Bars represents z-scored mean gene expression of each pathway, averaged across all ROIs in each hierarchy level. (b) Top: distribution of OXPHOS mean gene expression across the visual cortex shown for the left hemisphere. From left to right: lateral, medial, dorsal, ventral and posterior views. Colorbar represents the range of expression values within the visual cortex. Bottom: distribution of OXPHOS expression across individual visual ROIs. Bars represent z-scored mean expression of OXPHOS genes. dorsal, dorsal stream; ventral, ventral stream, MT+, MT+ complex and neighboring visual areas; ppp, pentose phosphate pathway; tca, tricarboxylic acid cycle; oxphos, oxidative phosphorylation; lactate, lactate metabolism and transport.

expression in the dorsal stream but lower expression in the primary visual cortex (V1). This is potentially in line with pre-vious reports of higher cytochrome oxidase reactivity in the magnocellular layers of the lateral geniculate nucleus relative to the parvocellular layers [62]. The dorsal stream is thought to be primarily driven by the magnocellular pathway, which processes high temporal frequencies, while the ventral stream is dominated by the parvocellular pathway that responds to low temporal frequency [62–65]. The differences in energy pathway expression may reflect the unique energy demands for visual attributes processed via the dorsal versus ventral streams. The PPP pathway, on the other hand, shows higher expression in the primary visual cortex. Together, these findings suggest that distinct components of the visual informa-tion processing hierarchy impose unique demands on energy metabolism, partially reflected in regional patterns of gene expression.

### Energy correlates of multi-scale cortical features

The heterogeneity of energy pathway profiles likely arises from the distinct molecular and cellular properties of different cortical regions at the microscale, as well as neurophysiological and network-level attributes at the macroscale. Corti-cal regions exhibit variability in their glucose uptake and oxygen consumption, as shown by molecular imaging. Neuro-physiological activity associated with signaling (i.e., action potentials and synaptic transmission) are energy-intensive and account for the majority of cortical energy demand [1,5,8,66]. Cellular specialization further shapes energy metabolism, as glial and neuronal populations exhibit distinct metabolic profiles [9]. Here, we characterize the spatial alignment of the energy pathways with a collection of maps corresponding to: (1) molecular imaging of metabolic uptake from PET [60], (2) neurophysiological oscillations from magnetoencephalography (MEG) [67], (3) cell type composition, (4) lam-inar organization [68,69] and (5) connectivity metrics (Fig 5). It should be noted that, unlike the PET and MEG maps which represent direct physiological measurements, cell type composition and laminar organization maps are derived from the AHBA using cell- and layer-specific gene markers as proxies for cell-type abundance and cortical laminar structure [68,70,71].

The cerebral metabolic rate of glucose ($CMR_{glc}$) PET map, which corresponds to glucose uptake, does not significantly correlate with glycolysis (rho = 0.22, $p_{spin,FDR}$ = 0.63) and OXPHOS maps (rho = 0.16, $p_{spin,FDR}$ = 0.76), however, it shows significant correlations with the lactate (rho = 0.53, $p_{spin,FDR}$ = 0.005) and PPP (rho = 0.42, $p_{spin,FDR}$ = 0.02) maps. Given that the PPP is generally characterized by minimal flux relative to other energy-producing pathways [6,13,48], this strong correlation with $CMR_{glc}$ is unexpected. However, it has been shown that glucose flux through the PPP can be largely underestimated [72,73]. Also, contrary to what we expected, the cerebral metabolic rate of oxygen ($CMR_{O2}$) map, repre-senting oxygen consumption in the brain, does not correlate with the OXPHOS map (rho = −0.01, $p_{spin,FDR}$ = 0.95, see *Discussion*).

There is a moderate alignment between $CMR_{O2}$ and the PPP map (rho = 0.54, $p_{spin,FDR}$ = 0.002), potentially highlight-ing greater need for cellular repair processes in regions with greater oxygen consumption and consequently higher oxida-tive damage. The Glycolytic Index shows significant positive correlations with the glycolysis (rho = 0.49, $p_{spin,FDR}$ = 0.047) and lactate (rho = 0.53, $p_{spin,FDR}$ = 0.04) maps, in line with the Glycolytic Index being a measure of aerobic glycolysis in the brain [60,74]. Unexpectedly, the PPP map shows no correlation with the Glycolytic Index map, despite contributing to non-oxidative glucose consumption. This may suggest that the PPP is not a major determinant of Glycolytic Index and that lactate production primarily accounts for aerobic glycolysis in the adult brain.

Furthermore, comparison with interpolated *ex vivo* maps of mitochondrial activity profile [75] reveals spatial alignment with gene expression-based energy maps. While TCA and OXPHOS maps do not align with mitochondrial density, they correlate significantly with mitochondrial respiratory capacity, which reflects mitochondrial specialization across brain regions (S5 Fig).

A multitude of steps bridge gene transcription and metabolic activity. This can be seen in the gene-wise spatial align-ment with $CMR_{glc}$, which reveals a wide range of correlations within each energy pathway (Fig 5B). Collectively, these

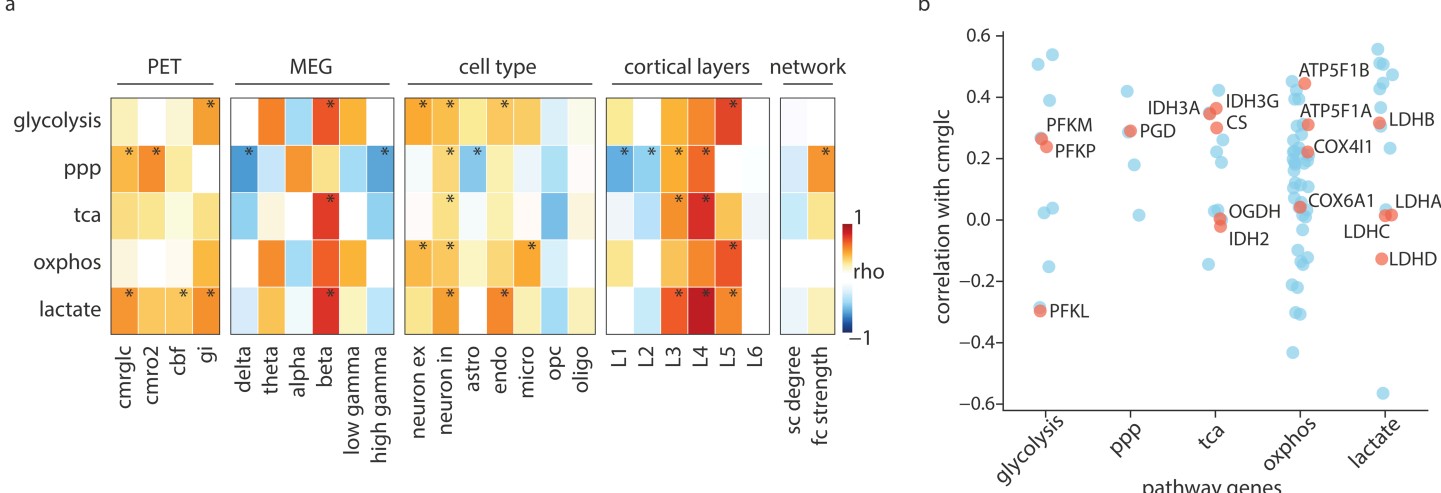

**Fig 5. Spatial correspondence of energy maps with multi-scale cortical annotations.** (a) Spearman's correlation between energy maps and *in vivo* metabolic PET imaging [60], neurophysiological oscillations [67], cell type composition, cortical laminar thickness [68] and network connectivity attributes [76]. Colors depict the strength of the correlation between pairs of brain maps. Asterisks indicate statistical significance when tested against a distribution of 10 000 spatial-autocorrelation preserving nulls after FDR-correction using the Benjamini-Hochberg method for multiple comparisons. (b) Correlation between individual genes in the energy pathways and PET-derived glucose uptake. Spearman's correlation was calculated between gene expression values and the $CMR_{glc}$ map. Red dots indicate genes encoding rate-limiting and key catalytic enzymes. $CMR_{glc}$, cerebral metabolic rate of glucose; $CMR_{O2}$, cerebral metabolic rate of oxygen; gi, glycolytic index; cbf, cerebral blood flow. Cell types: astro, astrocyte; neuron ex, excitatory neuron; neuron in, inhibitory neuron; endo, endothelial cell; micro, microglia; opc, oligodendrocyte precursor cells; oligo, oligodendrocyte; sc, structural connectivity; fc: functional connectivity.

results highlight the complexity of downstream metabolic pathways following glucose uptake, perhaps capturing different aspects of the underlying biology.

To examine how oscillatory activity is supported by energy metabolism pathways, we looked at the correspondence between energy maps and the six canonical MEG power bands. The energy maps primarily correlate with the beta band, reflecting their greater expression in the motor cortex (glycolysis: rho = 0.69, $p_{spin,FDR}$ = 0.03; OXPHOS: rho = 0.65, $p_{spin,FDR}$ = 0.07; TCA: rho = 0.74, $p_{spin,FDR}$ = 0.04; lactate: rho = 0.77, $p_{spin,FDR}$ = 0.005).

We next explored the correspondence between energy maps and cell type composition. Excitatory neurons show moderate correlations with the glycolysis (rho = 0.46, $p_{spin,FDR}$ = 0.001), and OXPHOS (rho = 0.42, $p_{spin,FDR}$ = 0.006) maps, in line with the higher energy demand of these principal cortical neurons [3,5,77]. Inhibitory neurons show positive correlations across all energy maps (glycolysis: rho = 0.42, $p_{spin,FDR}$ = 0.001; OXPHOS: rho = 0.37, $p_{spin,FDR}$ = 0.001; TCA: rho = 0.3, $p_{spin,FDR}$ = 0.003; Lactate: rho = 0.5, $p_{spin,FDR}$ = 0.001; PPP: rho = 0.33, $p_{spin,FDR}$ = 0.002). This may reflect both the greater energy demand and need for cellular repair processes due to their fast-spiking activity and susceptibility to oxidative damage. To test whether these associations can be attributed to the fast-spiking population, we further examined individual inhibitory neuron marker genes (S3 Table). Among inhibitory subtypes, parvalbumin expression shows a significant correlation with the PPP, TCA and lactate maps (S6 Fig), suggesting that the metabolic signature we observe may reflect the unique energy demands of fast-spiking parvalbumin-positive neurons. Endothelial cells show significant positive correlations with the glycolysis (rho = 0.38, $p_{spin,FDR}$ = 0.014) and lactate maps (rho = 0.56, $p_{spin,FDR}$ = 0.001), in line with the proposed glycolytic nature of these cells [78,79]. Microglia significantly correlate with the OXPHOS map (rho = 0.47, $p_{spin,FDR}$ = 0.003), consistent with their reliance on oxidative metabolism [2,80,81]. The PPP map however, exhibits a negative correlation with the astrocyte map and no significant correlation with microglia, OPC and oligodendrocytes, contrary to evidence of higher activity of this pathway in glial cells [13,82,83].

Given the diverse cellular composition and the distinct circuitry of cortical layers, we next sought to investigate the energy metabolism profile of cortical laminar organization. The PPP, TCA and lactate maps show the greatest alignment with cortical layer 4 (PPP: rho = 0.64, $p_{spin,FDR}$ = 0.001), TCA: rho = 0.79, $p_{spin,FDR}$ = 0.001, lactate: rho = 0.87, $p_{spin,FDR}$ = 0.001), potentially hinting at a greater need for fast energy supply and cellular biosynthesis in this sensory input layer. The glycolysis and the OXPHOS maps show significant alignment with cortical layer 5 (glycolysis: rho = 0.73, $p_{spin,FDR}$ = 0.001, OXPHOS: rho = 0.69, $p_{spin,FDR}$ = 0.001). Specifically, individual markers of excitatory layer 5 and Betz cells show significant correlations with glycolysis and OXPHOS (S6 Fig). This could hint to the greater metabolic demand of large pyramidal cells with their extensive subcortical projections [84,85].

Overall, energy maps exhibit diverse alignments across the different cell types and cortical laminar organization, pointing to the underlying compartmentalization of energy metabolism.

The relationship between energy metabolism and network connectivity can offer insights into the metabolic demands of topological hubs within cortical networks. Here, we find that the FC strength correlates positively with the PPP map (rho = 0.52, $p_{spin}$ = 0.015), suggesting greater anabolic demand in the functional hubs and in line with the higher Glycolytic Index and plasticity of these hub regions [74,86]. However, the FC strength does not correlate with either glycolysis or OXPHOS maps, contrary to existing literature reporting greater glucose uptake of functional hubs [87].

## Energy pathways track developmental milestones

Energy metabolic pathways are tightly coupled to nutrient availability and exhibit adaptive changes across the lifespan, underlying their integral role in supporting neurogenesis and synaptic growth and integrity. To investigate the developmental trajectory of energy pathways, we used the BrainSpan RNA-sequencing data [88]. As before, energy pathway gene sets were used to retrieve pathway-specific sample-by-gene expression matrices and expression was averaged across genes (S5 Table). We also investigated the expression trajectory of genes involved in ketone body utilization, given their importance as an obligate energy substrate during early development [20].

Fig 6 illustrates the distinct trajectories of these energy pathways. Glycolysis, TCA, OXPHOS, and lactate metabolism exhibit a similar trajectory, rising from the fetal stage to infancy, with OXPHOS peaking in childhood and subsequently showing a decline into the adolescent and adult levels. This is in accordance with previous reports of glucose and oxygen uptake peaking in childhood, followed by a decrease in oxidative pathway activity by adolescence [13,20]. This trend further resembles the trajectory of synaptic development genes (S7 Fig). The PPP pathway shows a sharp decline from the fetal stage to infancy, followed by a gradual decrease into adolescence and adulthood. This underscores the importance of the PPP in supporting brain tissue generation during early development [89–91] and closely aligns with the lifespan trajectory of neural progenitor cells (S7 Fig). Furthermore, we find that the average expression of genes involved in ketone body utilization increases from the fetal stage to infancy, before declining in early childhood, consistent with the postnatal trajectory of ketone body utilization during the nursing and weaning periods [4,20,92].

Region-wise analyses further show that these lifespan trajectories are consistent across cortical structures (S8 Fig). Collectively, these findings reveal the unique expression trajectories of energy metabolic pathways across the lifespan, offering complementary insight into the metabolic dynamics shaped by developmental needs and nutrient availability.

## Sensitivity analysis

As a final step, to ensure that the results are not influenced by preprocessing and analytic choices, we employ several sensitivity measures. First, given that only two out of six donors in the AHBA include samples from the right hemisphere, we reproduce the energy maps using expression data exclusively from the left hemisphere. Maps derived from the left hemisphere correlate significantly with those produced by mirroring across hemispheres (S9A Fig). Second, we reproduce the maps and repeat the analyses with the lower resolution Schaefer-100 parcellation (S9B Fig). Third, we repeat the analysis including all available genes regardless of differential stability (S6 Table). The spatial distribution of energy

Lifespan trajectory of energy pathway gene expression

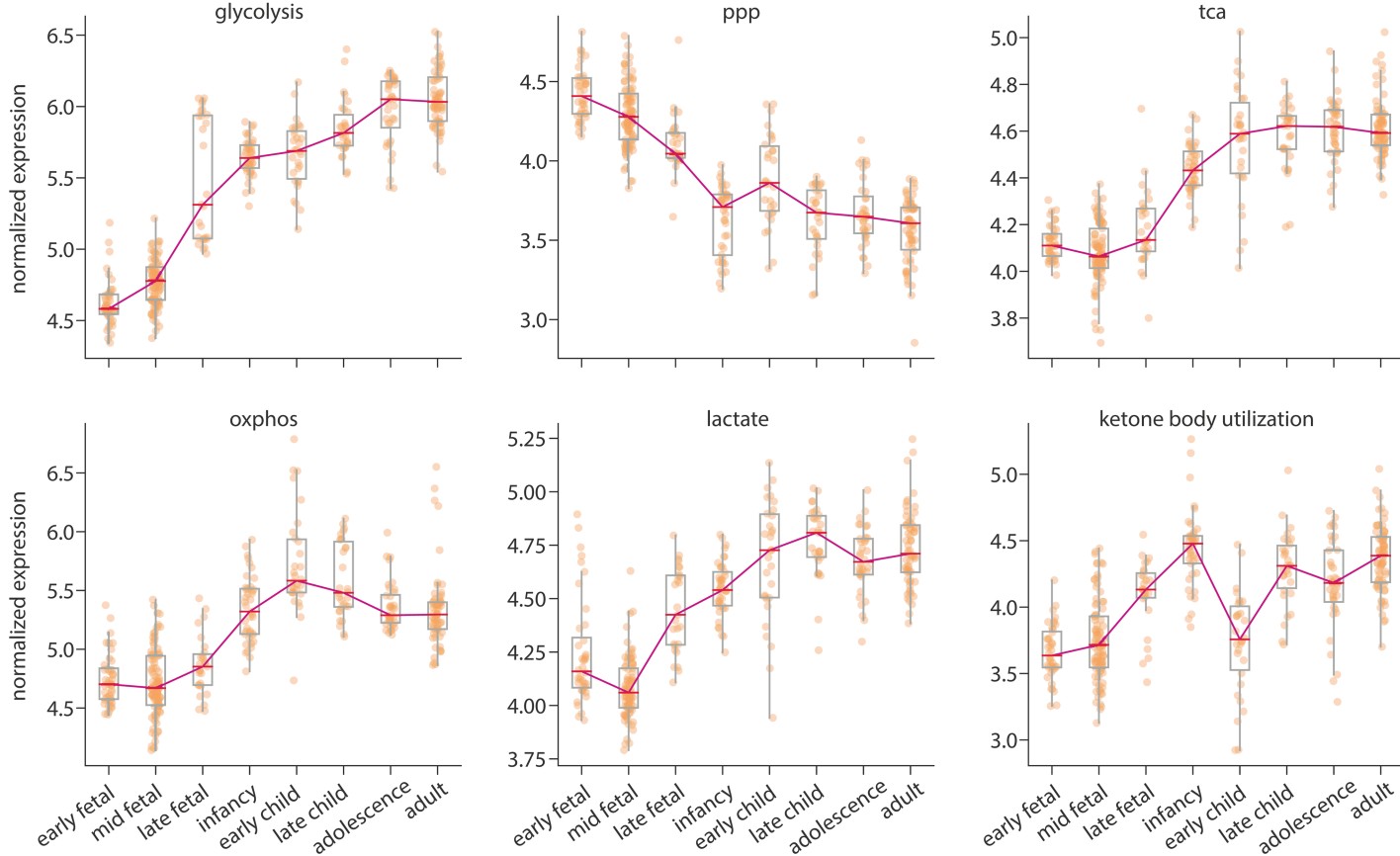

**Fig 6. Lifespan trajectory of energy pathway gene expression.** Developmental trajectories of energy metabolism pathways were produced using the BrainSpan RNA-sequencing dataset [88]. For each energy pathway, mean expression was calculated across all genes for each sample. Samples were grouped into major developmental stages and median expression across all samples falling into each age group was then calculated. Analysis only included cortical regions. The y-axis represents normalized $\log_2$(RPKM) expression values (see *Methods*. Dots represent individual samples. Line plot depicts the trajectory of median gene expression across age groups. For details of ages included in each group see S4 Table. ppp, pentose phosphate pathway; tca, tricarboxylic acid cycle; oxphos, oxidative phosphorylation; lactate, lactate metabolism and transport.

pathways remains consistent with the original analysis (S10 Fig). Fourth, to evaluate the robustness of our results to the choice of summary measure, we repeated the analysis using the first principal component (PC1) of gene expression for each pathway. The mean and PC1 maps are highly correlated and show consistent alignment with multi-scale cortical features (S11 Fig). Furthermore, subject-level analysis of these pathways shows that despite weak regional correlations across the 400 parcels (S12 Fig), enrichment within the cytoarchitectonic classes remains consistent across the six post-mortem brains, highlighting the divergence between the motor and sensory cortices (S13 Fig). We repeat the lifespan analysis using the non-parametric locally-estimated scatterplot smoothing (LOESS) method against log(age) (S14 Fig), modeling the trajectories without being constrained to predefined age groups. Lastly, we replicate the lifespan analysis using the BrainSpan microarray dataset [93] (S8 Table). Overall, the lifespan trajectories remain consistent with the original results (S15 Fig).

Finally, in addition to these five maps of key energy pathways, we also produced a complementary set of maps corresponding to other energy-related metabolic processes in the brain. These extended maps include: individual maps for the

five mitochondrial complexes, ketone body utilization, fatty acid metabolism, glycogen metabolism, branched chain amino acid (BCAA) catabolism, pyruvate dehydrogenase complex (PDC; responsible for the entry of pyruvate to the TCA cycle), malate-aspartate shuttle (MAS), glycerol phosphate shuttle (GPS), creatine kinase activity, detoxification of reactive oxygen species (ROS detox), generation of reactive oxygen species (ROS gen), nitric oxide signaling via guanylate cyclase (associated with vasodilation), $Na^+/K^+$ ATPase pump and the glutamine-glutamate cycle (S16 Fig and S9 Table for corresponding gene sets). S17 Fig shows the pair-wise correlation and the clustering among all the main and extended set of energy maps. For the lifespan trajectories of these extended energy processes see S19 Fig. Notably, mitochondrial ETC complexes show strong alignment with each other both in their spatial distribution across the cortex (rh̄o = 0.68, S18 Fig) and their lifespan trajectories (S19 Fig), reflecting their structural and functional coupling within the ETC. The greatest correlations are observed between complex I, III and IV, which reportedly form the most common supercomplexes. In contrast, complex II shows weaker correlations with the other complexes, which can be attributed to its less frequent incorporation in supercomplex formations, as well as involvement in the TCA [94–96].

## Discussion

We used a gene expression atlas to produce maps of five key energy pathways in the human brain. We observe a consistent dichotomy between glucose utilization for energy production versus cellular anabolism and antioxidant defense, which aligns with aspects of the cortical information processing hierarchy. Furthermore, we show that the developmental trajectories of these pathways follow critical milestones, reflecting shifts in metabolic requirements and tissue generation across the lifespan. Finally, we replicate our results using alternative processing approaches and datasets to examine the robustness of these findings. This study leverages pathway gene expression as a complementary approach to elucidate the biochemical blueprint of energy metabolism in the human brain.

We find that energy pathways show heterogeneous patterns of gene expression across the cortex with enrichment in specific structural and functional networks. The glycolysis and OXPHOS maps show higher expression in the primary motor cortices and lower expression in the visual cortex, hinting at differences in baseline energy requirements between these regions. The primary motor cortex, which generates efferent signals via large pyramidal neurons (Betz cells), demands substantial energy to support long-range axonal projections. On the other hand, the visual cortex is specialized for processing sensory input, has mostly short-range projections, and may rely on efficient encoding strategies and energy use. Interestingly, the creatine kinase (CK) map shows greater expression in the visual cortex (S16 Fig). CK catalyzes the reversible interconversion of creatine and phosphocreatine by transferring a phosphate group between ATP and ADP. The creatine-phosphocreatine shuttle therefore allows for fast regeneration of the ATP pool by continuous delivery of phosphate. CK is highly expressed in cells with high energy demand and is associated with sites of ATP production and consumption, such as mitochondria and ATPases (i.e. the sodium/potassium pump) [97,98]. Together with the greater expression of ATPase pump components (S16 Fig) in the visual cortex, this allows for a rapidly mobilizable pool of ATP, coupling substrate oxidation to the creatine-phosphocreatine shuttle and bypassing the electron transport chain, simultaneously reducing ROS production in mitochondria and exerting an indirect antioxidant effect [97].

The PPP displays greater expression in primary sensory cortices. This pathway is involved in anabolic tissue building processes and antioxidant defense [14,20,99]. The higher expression of PPP in primary sensory cortices may potentially reflect the greater demand for robust sensory coding [53], necessitating active maintenance of synaptic integrity through ongoing cellular biosynthesis. The PPP map also aligns with the T1w/T2w ratio and gene PC1 gradient maps (S20 Fig). The T1w/T2w map reflects the cortical hierarchical organization in the human brain and captures the cyto- and myelo-architectural boundaries [71]. The gene PC1 gradient represents the principal axis of transcriptional variation across the cortex: it follows the cortical hierarchical organization and differentiates the primary sensory and motor cortices from higher order association areas. This is in accordance with the role of the PPP in tissue building and synthesis of fatty acids and cholesterol essential for myelin production and further underscores PPP as a fundamental component of

brain microstructure and organization [90,100,101]. These spatial relationships are also reflected in the correspondence between energy maps and the first and second FC gradients. FC gradients represent the dominant differences in connectivity patterns across the cortex, which recapitulates the unimodal-transmodal hierarchy [54]. Glycolysis and OXPHOS align with the second functional gradient, which differentiates between the unimodal cortices. In contrast, the PPP and TCA capture the cortical hierarchy defined by the first functional gradient, separating primary regions from higher-order association and limbic cortices.

To examine how energy pathway gene expressions align with *in vivo* metabolic activity, we compared them to average maps from metabolic PET imaging. We find that the glycolysis and OXPHOS maps do not overlap significantly with resting-state glucose and oxygen consumption or cerebral blood flow measured by PET. The weak correspondence between OXPHOS gene expression and $CMR_{O2}$ likely reflects several factors. Mitochondrial proteins are among the most long-lived within the cell, allowing sustained function even after transcript depletion [102,103]. As a result, OXPHOS gene expression may better capture long-term metabolic capacity rather than acute oxygen use. In addition, $CMR_{O2}$ PET reflects resting oxygen consumption and does not account for spare respiratory capacity, which may be better indexed by transcriptional profiles. In contrast, the PPP shows the greatest correspondence with the $CMR_{O2}$ map. The overlap between PPP and $CMR_{O2}$ may indicate the need for cellular repair mechanisms in regions with greater oxygen consumption, which are more susceptible to oxidative damage. Indeed, the PPP is recruited by oxygen supplementation in newborn mice [104]. Conversely, the lactate map correlates most positively with the $CMR_{glc}$ and Glycolytic Index PET maps, in line with the contribution of lactate to non-oxidative glucose use [9,11,12,105,106]. Finally, it should be noted that individual genes within the energy pathways exhibit a wide range of correlations with $CMR_{glc}$, underscoring the existence of multiple intermediate steps between gene transcription and pathway activity. Moreover, FDG PET measures the first step in glycolysis, catalyzed by hexokinase, while our maps incorporate the expression of all the genes in each pathway.

Electrical activity in the brain relies on substantial energy [5,107]. We find that glycolysis, TCA and lactate maps most strongly overlap with the MEG beta band. Beta oscillatory activity is associated with task activation and motor processes and is thought to arise from GABAergic interneurons and bursting pyramidal cells [108]. The association of these three energy-producing pathways with beta power may underpin rapid energy supply via the astrocyte-neuron lactate shuttle during task-related neuronal activation [11,105,109].

Different cell types in the brain have distinct yet complementary metabolic gene expression and enzyme activity profiles [110,111]. Here, we show that excitatory neurons spatially align with the glycolysis and OXPHOS maps, in line with reports that glutamatergic signaling accounts for the majority of energy expenditure in the brain [3,5,112]. We also find that inhibitory neurons show spatial overlap with all energy pathways. Specifically, parvalbumin expression, the canonical marker of fast-spiking interneurons, shows pronounced alignment with the PPP, TCA and lactate maps (S6 Fig). The high-frequency spiking activity of parvalbumin-positive neurons imposes an energetic load on the brain, evidenced by the abundance of mitochondria in these cells [113–116], which renders this cell population especially vulnerable to oxidative damage [117,118]. This alignment likely reflects the recruitment of both energy-producing and anabolic pathways to support metabolic and cellular repair processes in these cells.

We also compared our energy pathway maps to spatial patterns of cortical laminar organization. The PPP, TCA and lactate maps exhibit positive correlations with cortical layers 3 and 4. Granular layer 4 is the main input layer of the cortex, characterized by high vascular density [119,120]. The three related energy pathways provide rapid energy supply upon the influx of sensory information necessary for accurate sensory coding. Conversely, glycolysis and OXPHOS spatially align with infragranular layer 5, the main output layer of the cortex, reflecting the greater static energy requirements of large pyramidal neurons such as Betz cells (S6 Fig) [84,85,121,122].

Regarding the relationship between energy metabolism and cortical network topology, we find that node degree does not show a correlation with glycolysis or OXPHOS maps, in contrast to previous studies reporting greater glucose uptake in high-degree nodes of the functional connectivity network [61,87]. This may indicate energy efficiency in these hub

regions, as the relationship between glucose uptake and FC degree appears to be nonlinear [87], suggesting efficient glucose utilization. On the other hand, the alignment between FC strength and the PPP supports the theory that hub regions require elevated anabolic activity to support neuronal plasticity [74,123,124].

Finally, we explore how energy metabolic pathways change throughout the human lifespan. Pathways primarily involved in ATP production, including glycolysis, TCA, OXPHOS and lactate metabolism, show a marked increase in expression from the fetal stage to infancy and peak in childhood. This aligns with research indicating that cerebral glucose and oxygen consumption, initially low at birth, rise rapidly postnatally [20,125,126]. In addition, OXPHOS expression decreases in adolescence and adulthood, in alignment with reports that oxygen consumption and the activity of oxidative pathways decline in adolescence [20,127,128]. This pattern is consistent across individual mitochondrial complexes (S19 Fig) and resembles the expression trajectory of synapse development markers (S7 Fig) and the normative trajectory of total cortical and grey matter volume [129–132]. Additionally, genes involved in ROS detoxification exhibit reduced expression into adulthood (S19 Fig). This could reflect diminished cellular defenses against oxidative stress in later stages of life [133]. The PPP shows a steep postnatal decrease, followed by a gradual decline in later life, consistent with its integral role in tissue generation for fetal brain development [13,20,90,134,135]. It has been demonstrated that PPP activity progressively declines with age and is undetectable in the rat brain by 18 months of age [135]. Furthermore, PPP is one of the major contributors to aerobic glycolysis. Aerobic glycolysis is considered a measure of non-oxidative glucose metabolism in the presence of oxygen [28,60,136,137], important for tissue biosynthesis [74,138,139]. Previous reports of decreased aerobic glycolysis in the aging brain may therefore be attributed to diminished PPP activity [140]. The PPP trajectory also resembles the rate of growth of mean cortical thickness across the lifespan, as well as the trajectories of genes involved in cell proliferation and neural progenitor cells (S7 Fig) [88,93,129]. Taken together, these findings underscore the critical role of PPP in providing anabolic support for brain tissue generation, emphasizing its importance for neurogenesis during fetal development [141] and waning influence with reduced plasticity in the aging brain [135,140,142]. Furthermore, given the role of PPP in antioxidant defense and repair, this decline could further underlie the reduced ROS buffering capacity and increased susceptibility to oxidative stress in later life [133,143,144].

Regarding ketone body utilization, we see a sharp increase from the fetal stage to infancy, followed by a decline in early childhood, mirroring the respective availability and use of ketone bodies during these developmental stages (Figs 6 and S15). In the adult brain, ketone bodies are mainly a source of fuel during starvation, but in the infant brain they serve a critical role both as an energy source and a substrate for synthesis of brain lipids [4,92]. During the nursing period, the high fat content of maternal milk results in elevated plasma concentrations of ketone bodies, making it an obligate fuel for the infant brain [145]. As weaning progresses and circulating ketone body levels decline, the brain shifts to glucose as fuel [20,128,128,146,147]. Here, we demonstrate that this distinct post-natal pattern is also present in the expression of genes involved in ketone body utilization. The subsequent increase in late childhood could be associated with adiposity increase at this stage [148,149] or point to a ketogenic shift in later life [150–152].

The present findings should be interpreted in light of several limitations. First, the gene expression data were obtained from only six post-mortem brains. While the brain transcriptome remains relatively stable with respect to post-mortem intervals, whole blood transcriptome suggests hypoxia-induced shifts in metabolism [153], which likely occur after death and may impact the generalizability and reliability of gene expression maps, especially ones based on energy metabolism. Furthermore, the AHBA samples were obtained from adults (ages 24–57, $42.50 \pm 13.38$) when the expression of energy metabolism genes and specifically the PPP genes has already declined. Additionally, some genes involved in energy metabolic pathways were unavailable or excluded based on quality control criteria (e.g., mitochondrial DNA genes that encode critical subunits of the electron transport chain were not probed in the AHBA and may exhibit distinct spatial patterns from nuclear-encoded genes due to different regulatory mechanisms). Nonetheless, the AHBA remains the most comprehensive high-resolution transcriptomic atlas of the human brain. Importantly, we see consistent results when restricting the analysis to the more extensively-sampled left hemisphere and when recreating the maps without

applying the differential stability threshold, as well as using a lower-resolution parcellation. Second, energy metabolism is characterized by interconnected pathways that branch out and converge via shared enzymes and metabolites [9]. Studying these pathways in isolation and relying on average expression is an oversimplification of their complex dynamics; nonetheless, we show consistent findings using PC1 as an alternative summary measure. Third, data used in the lifespan analyses are obtained from a different cohort and constitute a limited number of samples at each developmental stage, affecting the generalizability of the results. However, we show that the results remain consistent across both RNA-sequencing and microarray techniques. Finally, transcript levels do not directly correspond to pathway activity or metabolic flux, due to intervening regulatory steps (e.g., transcript splicing, post-translational modifications, protein ubiquitination, phosphorylation and degradation, enzymatic regulation). Therefore, these maps do not account for the post-translational modifications that underlie metabolic activity and specialization [9]. Moreover, the AHBA is obtained from bulk tissue samples where regional gene expression originates from mixed cell populations, limiting the ability to attribute specific energy profiles to a single cell type at this scale. Gene expression as used here could therefore be best viewed as the molecular blueprint of metabolic capacity.

Taken together, our results suggest that gene expression can be used to study the metabolic makeup of the human brain and its dynamics across the lifespan. We demonstrate the heterogeneous spatial distribution of key components of energy metabolism. We show that these energy pathways show distinct alignment across multiple scales of cortical organization and exhibit dynamic trajectories across the lifespan. The maps generated here provide a complementary perspective on the complexity and organization of brain energy metabolism beyond existing PET data, and add to the large corpus of normative anatomical and functional brain maps for neuroscience [154]. The energy pathway maps, along with the data and scripts used in this study, are made publicly available.

## Methods

### Energy pathway gene sets

Gene sets pertaining to energy metabolism pathways were curated using Gene Ontology (GO) [42] and Reactome Knowledge base [43]. These databases provide a unified platform for evidence-based and cross-referenced gene functional classification and pathway annotation and facilitate the study of biological systems, including genes, proteins and their interactions within a living organism. Pathways included in the main analysis are: glycolysis, pentose phosphate pathway (PPP), tricaboxylic acid cycle (TCA), oxidative phosphorylation (OXPHOS) and lactate metabolism and transport. For each of these pathways, we identified the corresponding GO biological processes [42] and Reactome pathway [43] IDs. We then retrieved gene sets involved in each pathways using *biomaRt* version 2.50.3 ([155], https://bioconductor. org/packages/release/bioc/html/biomaRt.html and *GO.db* packages (https://bioconductor.org/packages/GO.db/). *BioMart* is a freely available data-mining tool that provides unified access to biological knowledge bases. We used the Ensemble human gene annotation database release 112 [156] to retrieve gene sets for each pathway ID. Pathway gene sets retrieved from GO and Reactome databases are provided on our GitHub repository (https://github.com/netneurolab/ pourmajidian_metabolism-genes/). For each pathway, genes consistently annotated in both databases were retained for further analysis. Of the three hexokinase enzymes, only hexokinase 2 met the differential stability criterion (see next section). However, since hexokinase catalyzes the first step of both glycolysis and the PPP and therefore entrance to both pathways, it was excluded from these gene sets.

To provide a more comprehensive view of energy metabolism in the brain, we also produced an extended set of maps including: individual maps for the five mitochondrial complexes, ketone body utilization, fatty acid metabolism, glycogen metabolism, branched chain amino acid catabolism, pyruvate dehydrogenase complex (PDC; responsible for the entry of pyruvate to the TCA cycle), malate-aspartate shuttle (MAS), glycerol phosphate shuttle (GPS), creatine kinase (CK), detoxification of reactive oxygen species (ROS detox), generation of reactive oxygen species (ROS gen), the

glutamine-glutamate cycle, nitric oxide signaling and Na$^+$/K$^+$ ATPase pump. MAS, GPS, CK and the glutamine-glutamate cycle gene sets were further curated based on existing literature [20,98,157–159].

## Microarray gene expression data

Regional microarray expression data were obtained from 6 post-mortem brains (1 female, ages 24–57, $42.50 \pm 13.38$; postmortem interval 10–30 hours) provided by the Allen Human Brain Atlas (https://human.brain-map.org. All approvals and consent procedures can be found in the Allen Human Brain Atlas white paper and [41]. Data were processed using the Schaefer 400-region volumetric atlas in MNI space as described below. Microarray probes were reannotated using data provided by [160]; probes not matched to a valid Entrez ID were discarded. Next, probes were filtered based on their expression intensity relative to background noise [161], such that probes with intensity less than the background in $\geq 50\%$ of samples across donors were discarded, yielding 31 569 probes. When multiple probes indexed the expression of the same gene, we selected and used the probe with the most consistent pattern of regional variation across donors (i.e., differential stability; [162]).

MNI coordinates of tissue samples were updated to those generated via non-linear registration using the Advanced Normalization Tools (ANTs; https://github.com/chrisfilo/alleninf). To increase spatial coverage, tissue samples were mirrored bilaterally across the left and right hemispheres [163]. Samples were assigned to brain regions in the provided atlas if their MNI coordinates were within 2 mm of a given parcel. If a brain region was not assigned a tissue sample based on the above procedure, every voxel in the region was mapped to the nearest tissue sample from the donor in order to generate a dense, interpolated expression map. The average of these expression values was taken across all voxels in the region, weighted by the distance between each voxel and the sample mapped to it, in order to obtain an estimate of the parcellated expression values for the missing region. All tissue samples not assigned to a brain region in the provided atlas were discarded. Inter-subject variation was addressed by normalizing tissue sample expression values across genes using a robust sigmoid function [164]. Normalized expression values were then rescaled to the unit interval [44]. Gene expression values were then normalized across tissue samples using an identical procedure. Samples assigned to the same brain region were averaged separately for each donor, yielding a regional expression matrix for each donor with 400 rows, corresponding to the cortical regions in the Schaefer-400 parcellation, and 15 633 columns, corresponding to the retained genes. From this initial expression matrix, we retained genes with a differential stability value greater than 0.1 [71], yielding expression data for a total of 8 687 genes.

Energy pathway gene sets were used to extract pathway-specific gene expression matrices. The final number of genes per pathway differs from the original gene sets, as some genes were not present in the AHBA dataset or were excluded based on the above-mentioned quality control criteria applied during preprocessing. The differential stability distribution for energy pathway genes is shown in (S21 Fig). For each pathway, expression values were averaged across all genes to yield a pathway mean gene expression map. Brain maps were plotted using the *surfplot* package [165,166].

## Spatial auto-correlation preserving nulls

The brain exhibits inherent spatial auto-correlation in both structural and functional measures. Data mapped onto the brain such as gene expression are not independent and identically distributed (i.i.d.), which is a common prerequisite for many statistical tests. Given this spatial autocorrelation, nearby voxels/regions are more likely to have similar values (e.g. similar gene expression) due to both biological and technical (e.g., image processing and smoothing) factors [167]. This can lead to inflated statistical values.

To account for this, various spatial permutation tests (spin tests) have been introduced [168–170]. Spin tests account for the spatial auto-correlation present in brain data by permuting voxels/parcels while maintaining the spatial structure and auto-correlation, therefore providing a more accurate framework for hypothesis testing. In this study, we used the spatial permutation test developed by [169] implemented in the netneurotools package (https://netneurotools.readthedocs.

io/). This method uses parcel centroid coordinates to produce rotations and ensure that there are no duplicate reassignments, providing a true null distribution. We refer to the non-parametric p-value calculated using spatial permutation testing as $p_{spin}$.

## Parcellations, structural classes and functional networks

We used a cortical parcellation developed by [40] which divides the cortical surface into 400 regions. This parcellation was generated using a gradient-weighted Markov Random Field model from resting state fMRI data, integrating both local gradients and global similarity to define parcel boundaries. To investigate how our energy maps are distributed across functional networks, we used the resting state network assignments provided by the original authors which is based on the seven intrinsic functional networks described by [58]. To explore networks pertaining to structural classes, we used two network definitions: von Economo-Koskinas cytoarchitectonic classes based on the morphology and laminar differentiation of neuronal types [50,52,171,172], and Mesulam classes describing the sensory-fugal hierarchy of information processing [53,173].

For analysis of energy expression within the visual cortex, we used the Glasser parcellation ([59], https://github.com/brainspaces/glasser360/). Functional delineation of the visual cortex hierarchy was obtained from [59] supplementary neuroanatomical results and https://neuroimaging-core-docs.readthedocs.io/en/latest/pages/atlases.html. Subcortical energy maps were produced using the Desikan-Killiany atlas [174] and plotted using the ENIGMA Toolbox [175].

## Functional connectivity gradients

The first two functional connectivity (FC) gradients calculated by Margulies et al. [54] were retrieved from the *neuromaps* package [176]. Briefly, gradients were calculated for 820 healthy individuals from the Human Connectome Project (HCP) S900 release. All experimental procedures and consent information for the HCP was approved by the Institutional Review Board at Washington University [177]. The affinity matrix was then calculated from the FC matrix using cosine distance. FC gradients were computed using diffusion embedding, a non-linear dimensionality reduction technique that projects the data into a low-dimensional embedding space and assures a more stable representation of the connections compared to other dimensionality reduction techniques [54]. FC gradients were parcellated according to the Schaefer-400 atlas [40].

## Metabolic PET neuroimaging data

Metabolic PET maps were produced previously in [60] and retrieved from the *neuromaps* package [176]. Ethical approvals and informed consent for the original study were obtained from the Human Research Protection Office and the Radioactive Drug Research Committee at Washington University in St. Louis.[60]. These PET maps include: $CMR_{glc}$ using [$^{18}F$]-labeled fluorodeoxyglucose (FDG) radiotracer, $CMR_{O2}$ and cerebral blood flow (CBF) using [$^{15}O$]oxygen and water. All PET maps were obtained from neurologically normal individuals ($n = 33$, 14 males; age $= 25.4 \pm 2.6$ years) at resting state, using a Siemens model 961 ECAT EXACT HR 47 PET scanner [60]. A map of Glycolytic Index from the same study was also included in the analysis. The Glycolytic Index map is produced using the residuals after linearly regressing $CMR_{glc}$ on $CMR_{O2}$ and it was introduced previously as a measure of non-oxidative metabolism of glucose (aerobic glycolysis) [60]. Metabolic PET maps were then parcellated into the Schaefer-400 parcellation using the *neuromaps* package.

## Magnetoecephalography maps

MEG frequency data were first processed and used in [67] and were retrieved using the *neuromaps* package [176]. Resting state MEG data of a set of healthy young adults (n =33, 17 males; age 22–35 years) with no familial relationships were obtained from HCP (S900 release; [177]). The data include resting state scans of about 6 minutes duration (sampling rate =2 034.5 Hz; anti-aliasing lowpass filter at 400 Hz) and noise recordings for all participants. The data was analyzed using BrainStorm [178]. Pre-processing was performed by applying notch filters at 60, 120, 180, 240 and 300 Hz and

was followed by a high-pass filter at 0.3 Hz to remove slow-wave and DC-offset artifacts. The artifacts (including heartbeats, eye blinks, saccades, muscle movements, and noisy segments) were then removed from the recordings using automatic procedures as proposed by Brainstorm. Pre-processed sensor-level data were used to obtain a source estimation on HCP's fsLR4k cortex surface for each participant. Head models were computed using overlapping spheres, and the data and noise covariance matrices were estimated from the resting-state MEG and noise recordings. Brainstorm's linearly constrained minimum variance beamformers method was applied to obtain the source activity for each participant. Data covariance regularization was performed using the "median eigenvalue" method from Brainstorm[178]. The estimated source variance was also normalized by the noise covariance matrix. Source orientations were constrained to be normal to the cortical surface at each of the 8004 vertex locations on the fsLR4k surface. Welch's method was then applied to estimate power spectrum density for the source-level data, using overlapping windows of length 4 seconds with 50% overlap. Average power at each frequency band was then calculated for each vertex as the mean power across the frequency range of a given frequency band. The power spectrum was computed at the vertex level across six canonical frequency bands: delta (2–4 Hz), theta (5–7 Hz), alpha (8–12 Hz), beta (15–29 Hz), low gamma (30–59 Hz) and high gamma (60–90 Hz). Group-averaged maps for each MEG frequency bands were retrieved from the *neuromaps* package [176] and parcellated according to the Schaefer-400 cortical atlas [40].

## Cell and layer specific gene expression maps

Brain cell type- and layer-specific maps were made using marker gene sets which were obtained from Wagstyl et al. [68], supplementary file 2. Briefly, the authors curated cell-type markers by combining data from multiple single-cell and single-nucleus RNA-sequencing studies [88,179–185]. Subcategories across these studies were grouped into seven canonical classes including: excitatory neurons, inhibitory neurons, astrocytes, endothelial cells, microglia, oligodendrocytes and oligodendrocyte progenitor cells. Cell type gene sets were used to filter the AHBA gene expression matrix to obtain region-by-gene cell-specific expression matrices. Expression was then averaged across genes to yield a cell type mean expression map corresponding to the Schaefer-400 parcellation. Layer specific gene sets were curated based on two RNA-sequencing studies using samples from the prefrontal cortex [186,187]. Layer-specific gene sets were combined in these two studies by Wasgtyl et al. [68]. Layer-specific mean expression maps were produced as above. Gene markers for inhibitory and excitatory neuronal subtypes were retrieved from [93] and [184].

## Functional and structural connectivity measures

Both functional magnetic resonance imaging (fMRI) and diffusion weighted imaging (DWI) data were previously obtained for 326 unrelated participants (145 males; age 22–35 years) from the Human Connectome Project (HCP) S900 release [76,177,188]. All data were anonymized, and the original study protocol was approved by the WU-Minn HCP Consortium [177]. fMRI data was acquired using a 3T scanner for 15 minutes during the resting state. All 4 resting state fMRI scans (2 scans with R/L and L/R phase encoding directions on day 1 and day 2, each about 15 minutes long; TR =720 ms) were available for all participants. Preprocessing was previously performed using the minimal preprocessing pipeline [189]. Briefly, all 3T functional MRI time-series (voxel resolution of 2mm isotropic) were corrected for gradient nonlinearity, head motion using a rigid body transformation, and geometric distortions using scan pairs with opposite phase encoding directions (R/L, L/R) [128]. Further preprocessing steps include coregistration of the corrected images to the T1w structural MR images, brain extraction, normalization of whole brain intensity, high-pass filtering (>2,000s FWHM; to correct for scanner drifts), and removing additional noise using the ICA-FIX process [76,190]. The fMRI time series were then parcellated into 400 cortical regions in the Schaefer atlas [40] and the functional connectivity matrix was generated by computing the Pearson's correlation coefficient between pairs of regional time series for each of the 4 scan each participant. A group-average functional connectivity matrix was computed representing mean functional connectivity across all subjects and normalized using a Fisher's r-to-z transformation [191,192].

Structural connectomes were previously generated from minimally processed HCP S900 DWI data using the MRtrix3 package [193]. Multi-shell and multi-tissue response functions were estimated and spherical-deconvolution informed filtering of tractograms (SIFT2) was applied to reconstruct whole brain streamlines weighted by cross-section multipliers [194,195]. The initial tractogram was generated with 40 million streamlines, with a maximum tract length of 250. For each subject, these reconstructed cross-section streamlines were mapped onto the Schaefer-400 atlas [40] to build a structural connectome. A group-consensus binary network was constructed, preserving the density and edge-length distributions of the individual connectomes [196].

Graph theory can be used to study the brain as a network of interconnected nodes and edges, where nodes represent distinct regions or units within the brain (i.e., neurons or parcels), and edges are the connections or interactions between these units (structural connection or functional co-activation between pairs of nodes) [197–199]. Graph theory therefore allows us to define measures of regional importance in the brain connectivity network. We used the *bctpy* package (https://github.com/aestrivex/bctpy) to calculate these network measures from the binary structural connectome and the functional connectivity matrix including degree centrality and strength [200]. **Degree** is defined as the number of edges (connections) of a node. It is a local network measure calculated at each node that represents "hubness" in a network. **Strength** is the weighted analogue of degree centrality. In a weighted matrix such as the functional connectivity matrix, strength is calculated as the sum of connection weights incident on a node.

## Mitochondrial phenotype maps

The mitochondrial phenotype maps including mitochondrial density, complex I, complex II and complex IV enzymatic activity, tissue respiratory capacity and mitochondrial respiratory capacity are publicly available and were obtained from [75] (https://neurovault.org/collections/16418/). The post-mortem brain tissue in this study was collected by the Macedonian-New York State Psychiatric Institute (NYSPI) Brain collection and made available through Quantitative Brain Biology Institute at the NYSPI. All approvals for the original study were obtained from the Institutional Review Board at NYSPI [75]. Briefly, a 2-cm-thick frozen coronal slab of the right hemisphere (54 years old, neurotypical male donor) was physically voxelized into 703 samples (3mm isotropic) using a computer-controlled cryo-milling technique. Mitochondrial density was indexed using mitochondrial DNA and citrate synthase. Measures of enzymatic activity were assessed using independent respirometry and colorimetry assays. Tissue respiratory capacity was defined as the average enzymatic activity of the three mitochondrial complexes. Mitochondrial respiratory capacity was defined as tissue respiratory capacity scaled by mitochondrial density. The 3mm resolution mitochondrial maps were then separately regressed onto 22 MRI-derived structural, functional, and diffusion metrics using stepwise linear regression. The model parameters were then used to produce whole brain maps of mitochondrial features using the same MRI-based metrics at the 1mm resolution [75].

## Human brain lifespan transcriptomics data

BrainSpan is a freely available database containing the developmental transcriptome of the human brain spanning the pre-natal stages to adulthood (https://www.BrainSpan.org/static/download.html/). Tissue was collected after obtaining consent and with approval from the institutional review boards of Yale University School of Medicine, the National Institutes of Health, and contributing institutions. Tissue handling was done in compliance with the NIH ethical guidelines and the WMA Declaration of Helsinki [88]. The data includes 524 samples from 42 donors (19 females) across 31 developmental stages spanning from 8 weeks post conception (PCW) to 40 years of age. The samples were taken from a total of 26 cortical and sub-cortical regions in the brain (BrainSpan white paper). RNA-sequencing data was previously processed into normalized Reads Per Kilobase of transcript per Million (RPKM) using conditional quantile normalization to account for GC content and sequencing depth and batch effect correction using ComBat [88,201].

RNA-sequencing data (Genecode v10 summarized to genes) were downloaded from the BrainSpan database (BrainSpan Download). The expression matrix contains normalized RPKM values for 52 376 genes across 524 samples.

We grouped the samples into major developmental stages including: early fetal, mid fetal, late fetal, infancy, early childhood, late childhood, adolescence and adulthood [93,202] (S4 Table). All subsequent analysis was performed on the cortical samples. First, We carried out a basic cleanup of the sample-by-gene expression matrix: (1) We retained regions that had at least 1 sample in each age group. (2) Duplicate genes were removed, yielding 47 808 unique genes. (3) Genes were retained if they had an RPKM value >= 1 in 80% of the samples at each spatiotemporal point [203].

The cleanup step resulted in a 352 samples from 11 cortical regions and 8 370 genes. Cortical regions include: rostral anterior cingulate, medial prefrontal cortex, dorsolateral prefrontal cortex, inferolateral temporal cortex (area TEv area 20), orbitofrontal cortex, posterior (caudal) superior temporal cortex (area 22c), posteroventral (inferior) parietal cortex, primary auditory cortex (core), primary motor cortex (M1, Brodmann area 4), primary somatosensory cortex (S1, areas 312), primary visual cortex (striate cortex V1, area 17) and ventrolateral prefrontal cortex.

Expression values were $log_2$ transformed and normalized using the upper quartile method [204–206]. Each donor's data was scaled by their 75th percentile expression value and multiplied by the mean 75th percentile value across all donors. We then retrieved sample-by-gene expression matrices for each energy pathway using the curated gene sets. Average expression across all genes for each energy pathway was calculated for each sample. Mean energy pathway gene expression was then aggregated into the eight age groups by combining all samples within each respective age group. Pathway expression was then plotted across age categories. Marker genes for neural progenitor cells and synapse development were obtained from the supplementary materials of [88,93]. Smoothed curves were produced using the LOESS method against $log_{10}(age)$ in post conception days using the *rpy2* package https://rpy2.github.io/doc.html. For the microarray dataset, genes were retained if they had a $log_2(expression) \geq 6$ [93] and the rest of the analysis was carried out as above.

## Supporting information

**S1 Fig. PGD gene expression correlates with glycolysis and OXPHOS maps.** Brain map depicts phosphogluconate dehydrogenase (*PGD*) gene expression according to the Schaefer-400 parcellation. Colorbar represents z-scored expression values. *PGD* expression was correlated (Spearman's) with glycolysis and OXPHOS mean expression maps. Correlations were tested against a distribution of 10 000 correlations produced from the spatial permutation testing. The non-parametric p-value is indicated as $p_{spin}$. Dots in the scatter plot represent 400 cortical regions in the Schaefer-400 parcellation.
(PDF)

**S2 Fig. Principal component analysis of energy pathway gene expression.** (a) Brain maps showing the first principal component (PC1) of pathway gene expression matrices. PC1 of the glycolysis and OXPHOS gene expression reflect a gradient from the motor and prefrontal cortices to the parietal association regions, and the visual cortex (glycolysis: $var_{explained} = \%45.46$; OXPHOS: $var_{explained} = \%55.01$). PC1 of the PPP gene expression shows a spatial pattern closely capturing the established global gene expression gradient, extending from the sensory cortices to the higher order association, and limbic areas [71] ($var_{explained} = \%32.30$). The PC1 maps can be found in S1 Data. (b) Percent of variance explained by the first five principal components of pathway gene expression. (c) Spearman's correlation between energy maps and the PC1 of expression of all genes in the AHBA. (d) Correlation between energy maps and average expression of all genes in the AHBA. Highlighted bars represent statistical significance when tested against a distribution of 10 000 spatial-autocorrelation preserving nulls ($p_{spin} < 0.05$). The correlation and $p_{spin}$ values can be found in S1 Data. ppp, pentose phosphate pathway; tca, tricarboxylic acid cycle; oxphos, oxidative phosphorylation; lactate, lactate metabolism and transport.
(PDF)

**S3 Fig. Subcortical energy pathway profiles.** Energy pathway gene expression matrices were retrieved for 14 subcortical regions in the Desikian-Killiany atlas [174]. Left: subcortical visualization of mean pathway gene expression. Ventricles are excluded due to the absence of gene expression data. Stable genes ($ds \geq 0.1$) were retained to produce pathway mean gene expression maps (see *Methods*). Colorbar represents expression values. Right: Barplot representation of the subcortical energy profiles (left hemisphere). Bars correspond to pathway mean gene expression, z-scored across all subcortical regions. Energy pathways consistently show higher expression in the thalamus and lower expression in the amygdala [55]. ppp, pentose phosphate pathway; tca, tricarboxylic acid cycle; oxphos, oxidative phosphorylation; lactate, lactate metabolism and transport.
(PDF)

**S4 Fig. Distribution of energy pathway maps across intrinsic functional networks.** Maps were z-scored across the 400 cortical regions and the average expression of parcels falling into each functional network was calculated for each energy map, according to the Yeo-Kiernen intrinsic functional network parcellation [58]. Highlighted bars indicate statistical significance when tested against 10 000 spatial-autocorrelation preserving nulls ($p_{\text{spin}} < 0.05$). Brain plots visualize parcels making up each functional network. Glycolysis, TCA, OXPHOS and lactate maps show significantly greater values in the somato-motor cortex (glycolysis: $p_{\text{spin}} = 0.049$; TCA: $p_{\text{spin}} = 0.0005$; OXPHOS: $p_{\text{spin}} = 0.03$; lactate: $p_{\text{spin}} = 0.02$). Glycolysis and OXPHOS have significantly lower expressions in the visual cortex (glycolysis: $p_{\text{spin}} = 0.003$; OXPHOS: $p_{\text{spin}} = 0.004$). The PPP map on the other hand shows greater expression in the visual cortex, although not significant when tested against spatial permutations ($p_{\text{spin}} = 0.09$) and significantly lower expression in the limbic network ($p_{\text{spin}} = 0.006$). Data underlying this figure can be found in S1 Data. ppp, pentose phosphate pathway; tca, tricarboxylic acid cycle; oxphos, oxidative phosphorylation; lactate, lactate metabolism and transport.
(PDF)

**S5 Fig. Correlation between energy gene expression and *ex-vivo* mitochondrial phenotype maps.** Mitochondrial phenotype maps were obtained from [75] and parcellated into Schaefer-400. The y-axis represents mitochondrial phenotype maps and the x-axis represents gene expression-based maps. Orange scatter plots indicate statistically significant correlations (Spearman's) tested against 10 000 spatial-autocorrelation preserving nulls ($p_{\text{spin}} < 0.05$). CI, mitochondrial complex 1 activity; CII, mitochondrial complex 2 activity; CIV, mitochondrial complex 4 activity; MitoD, mitochondrial density; TRC, tissue respiratory capacity; MRC, mitochondrial respiratory capacity; tca, tricarboxylic acid cycle; oxphos, oxidative phosphorylation.
(PDF)

**S6 Fig. Spatial alignment between energy maps and individual inhibitory and excitatory subtypes.** For each individual sub-type, gene markers were obtained from [93] and [184]. For the list of gene markers see S3 Table. (Left) Heatmap showing the strength of correlation between energy and cell-type maps. Colorbar indicates Spearman's correlation values. Asterisks indicate statistical significance when tested against a distribution of 10 000 spatial-autocorrelation preserving nulls after FDR-correction using the Benjamini-Hochberg method for multiple comparisons. (Right) Cortical distribution of neuronal subtype gene expression. Colorbar represents expression values across the 400 Schaefer regions. Cell types: pvalb, parvalbumin; sst, somatostatin; calb, calbindin; vip, vasoactive intestinal peptide; exc, excitatory.
(PDF)

**S7 Fig. Lifespan trajectory of genes related to neurodevelopmental processes.** Expression trajectory of neurodevelopmental processes were produced using previously curated gene sets [88,93]. Mean expression of marker genes for each developmental process was calculated for each sample. Samples were then grouped into age categories. Line plot represents median values across all samples in each age group. Analysis only included cortical regions. The y-axis repre-

sents upper quartile normalized $\log_2$(RPKM) values (see *Methods*). Dots represent individual samples in each age group. For details of ages included in each group see S4 Table.
(PDF)

**S8 Fig. Region-wise developmental trajectory of energy maps.** Lifespan analysis was done using the non-parametric locally estimated scatterplot smoothing (LOESS) method. For each energy pathway, mean expression was calculated across all genes for each sample. Samples were grouped into 11 cortical regions available in the dataset. The x-axis represent $\log_{10}$ transformed age in post conception days. Dots represent individual cortical samples at each age colored by region. The y-axis shows upper quartile normalized $\log_2$(RPKM) values. ppp, pentose phosphate pathway; tca, tricarboxylic acid cycle; oxphos, oxidative phosphorylation; lactate, lactate metabolism and transport.
(PDF)

**S9 Fig. Sensitivity analysis.** To assess the robustness of the results, energy maps were reproduced using (1) only the left hemisphere microarray data, and (2) a coarser parcellation (Schaefer-100). (a) Maps produced using only left hemisphere data and their correlation with the left hemisphere of maps in the main analysis (produced by mirroring across hemispheres). Scatter plot represents 200 regions in the left hemisphere parcellated according to Schaefer-400 parcellation. (b) Energy maps according to Schaefer-100 parcellation. Left: Heatmap depicts the pairwise correlation among all genes included in the energy sets across 100 regions in the Schaefer-100 parcellation. Middle: Spearman's correlation between mean expression energy maps. Right: correlation of energy maps with the first principal component of all genes in AHBA (gene PC1). Brain map color bars represent z-scored expression across all regions in each parcellation. Highlighted bars show statistical significance when tested against 1 000 spatial-autocorrelation preserving nulls. Data underlying this figure can be found in S1 Data. lh, left hemisphere; ppp, pentose phosphate pathway; tca, tricarboxylic acid cycle; oxphos, oxidative phosphorylation; lactate, lactate metabolism and transport.
(PDF)

**S10 Fig. Energy maps without the differential stability threshold.** Energy maps were reproduced using all available energy genes in the AHBA regardless of their differential stability threshold. (a) Spearman's correlation between $ds \geq 0.1$ maps and maps with no differential stability threshold. Correlations were tested against a distribution of 10 000 nulls produced from the spatial permutation testing. The non-parametric p-value is indicated as $p_{\text{spin}}$. Dots represent 400 cortical regions in the Schaefer-400 parcellation. (b) Left: Heatmap depicts Spearman's correlation between mean expression energy maps. Middle: correlation of energy maps with the first principal component of all genes in AHBA (gene pc1). Left: Alignment between the FC gradients and energy maps. (c) Top: enrichment of energy maps across the seven von Economo cytoacrhitectonics classes. Bottom: enrichment of energy maps across the Mesulam sensory-fugal axis of information processing. The y-axis represents mean gene expression of z-scored maps. Highlighted bars indicate statistical significance ($p_{\text{spin}} < 0.05$). Data for this figure is provided in S1 Data. ds, differential stability; ppp, pentose phosphate pathway; tca, tricarboxylic acid cycle; oxphos, oxidative phosphorylation; lactate, lactate metabolism and transport.
(PDF)

**S11 Fig. Energy maps produced from the first principal component of pathway gene expression.** (a) Alignment between the mean gene expression maps and the first principal component of gene expression for each energy pathway. Dots in the scatterplot represent 400 cortical regions in the Schaefer-400 parcellation. (b) Spatial alignment among energy PC1 maps calculated using pairwise Spearman's correlation. (c) Spearman's correlation of PC1 energy maps and multi-scale cortical features. Asterisk represents significance after permutation testing using 10 000 spatial nulls and FDR correction using the Benjamini-Hochberg method for multiple comparison. Colorbar shows Spearman's correlation values. $CMR_{glc}$, cerebral metabolic rate of glucose; $CMR_{O2}$, cerebral metabolic rate of oxygen; gi, glycolytic index; cbf, cerebral blood flow. Cell types: astro, astrocyte; neuron ex, excitatory neuron; neuron in, inhibitory neuron; endo, endothelial

cell; micro, microglia; opc, oligodendrocyte precursor cells; oligo, oligodendrocyte; sc, structural connectivity; fc: functional connectivity.
(PDF)

**S12 Fig. Correlation between subject level energy maps.** Gene expression matrices were retrieved for each subject using *abagen* package. Genes with $ds \geq 0.1$ were kept and pathway mean gene expression maps were produced as before (*See methods*. For each pathway, the heatmap depicts Spearman's correlation between individual subjects. The x- and y-axis corresponds to the donor IDs. ppp, pentose phosphate pathway; tca, tricarboxylic acid cycle; oxphos, oxidative phosphorylation; lactate, lactate metabolism and transport.
(PDF)

**S13 Fig. Distribution of subject-level energy gene expression across the von Economo classes.** Maps were z-scored across the 400 cortical regions and the average expression of parcels falling into each class was calculated for each energy pathway. The y-axis represents mean gene expression of z-scored maps. Highlighted bars indicate statistical significance when tested against 10 000 spatial-autocorrelation preserving nulls ($p_{spin} < 0.05$). Data for this figure can be found in S1 Data. ppp, pentose phosphate pathway; tca, tricarboxylic acid cycle; oxphos, oxidative phosphorylation; lactate, lactate metabolism and transport.
(PDF)

**S14 Fig. Lifespan trajectory of energy maps using LOESS.** Lifespan analysis was repeated using the non-parametric locally estimated scatterplot smoothing (LOESS) method. For each energy pathway, mean expression was calculated across all genes for each sample. The x-axis represent $\log_{10}$ transformed age in post conception days. Dots represent individual cortical samples at each age. The y-axis shows upper quartile normalized $\log_2$(RPKM) values. ppp, pentose phosphate pathway; tca, tricarboxylic acid cycle; oxphos, oxidative phosphorylation; lactate, lactate metabolism and transport.
(PDF)

**S15 Fig. Microarray expression trajectory of energy maps across the lifespan.** Lifespan analysis was repeated using the microarray data from the BrainSpan dataset [93]. For each energy pathway, mean expression was calculated across all genes for each sample. Samples were then grouped into eight age bins and the median expression across all samples falling into each age bin was calculated. Analysis only included cortical regions. The y-axis represents upper quartile normalized $\log_2$(signal intensity) (see *Methods*). Dots represent individual samples in each age group. Line plot depicts the trajectory of median gene expression. For details of ages included in each group see S8 Table. ppp, pentose phosphate pathway; tca, tricarboxylic acid cycle; oxphos, oxidative phosphorylation; lactate, lactate metabolism and transport.
(PDF)

**S16 Fig. Extended set of energy related pathways.** Mean gene expression maps were produced as before (see *Methods*). Color bars represent mean gene expression, z-scored across the 400 cortical regions in the Schaefer-400 parcellation. The clustering analysis of these maps can be found in S17 Fig. The lifespan trajectory of the extended set of energy maps can be found in S19 Fig. atpsynth, ATP synthase complex; BCAA, branch chained amino acid; pdc, pyruvate dehydrogenase complex; mas, malate-aspartate shuttle; gps: glycerol-3-phosphate shuttle; ros detox, detoxification of reactive oxygen species; ros gen, generation of reactive oxygen species; no signaling, nitric oxide signaling; gln-glu cycle, glutamine-glutamate cycle.
(PDF)

**S17 Fig. Clustering analysis of energy maps.** (a) Left: Clustering of the complete set of energy maps was carried out using PCA on a matrix containing all pathway mean gene expressions. Axes represent the first and second principal

components. Right: hierarchical clustering of extended energy maps. Colorbar represents expression values across the 400 cortical regions in the Schaefer parcellation. (b) Pair-wise correlation among the extended energy maps. Colorbar represents Spearman's correlation values. Bold edges indicate statistical significance tested against 10 000 spatial nulls and after FDR correction using the Bejamini-Hochberg method for multiple comparisons. atpsynth, ATP synthase complex; BCAA, branch-chained amino acids; pdc, pyruvate dehydrogenase complex; mas, malate-aspartate shuttle; gps: glycerol-3-phosphate shuttle; ros gen, generation of reactive oxygen species; no signaling, nitric oxide signaling; gln-glu cycle, glutamine-glutamate cycle.
(PDF)

**S18 Fig. Alignment between the components of mitochondrial respiratory chain.** Left: Venn diagram depicts the final number of genes in each mitochondrial complex and their overlap. Right: Heatmap depicts pairwise Spearman's correlations between mean gene expression maps. Gene sets for each mitochondrial complex were retrieved using GO pathway IDs and mean expression maps where produced as before (see *Methods*). Asterisks show statistical significance when tested against a distribution of 10 000 correlations produced using spatial autocorrelation preserving permutation test. atpsynth, ATP synthase complex.
(PDF)

**S19 Fig. Lifespan trajectory of the extended energy metabolism pathways.** Developmental trajectories of energy metabolism pathways were produced using the BrainSpan RNA-seq expression dataset. For each energy pathway, mean expression was calculated across all genes for each sample and median expression across all samples falling into each age group was calculated. Analysis only included cortical regions. the y-axis represents upper quartile normalized $\log_2$(RPKM) expression values. Dots represent individual samples in each age group. Line plot depicts the trajectory of median gene expression. atpsynth, ATP synthase complex; FA, fatty acid; BCAA, branch chained amino acid; pdc, pyruvate dehydrogenase complex; mas, malate-aspartate shuttle; gps: glycerol-3-phosphate shuttle; ros detox: detoxification of reactive oxygen species; ros gen, generation of reactive oxygen species; no signaling, nitric oxide signaling; gln-glu cycle, glutamine-glutamate cycle.
(PDF)

**S20 Fig. Alignment between the PPP and cortical T1w/T2w map.** Spatial alignment between the maps were calculated using the Spearman's correlation and tested against a distribution of 10 000 correlations produced from the spatial permutation nulls. The x-axis represents the PPP mean gene expression. Dots in each scatter plot represent the 400 cortical regions in the Schaefer-400 parcellation. ppp, pentose phosphate pathway.
(PDF)

**S21 Fig. Differential stability of energy pathway genes.** The differential stability distribution of the final energy pathway genes. Expression data were filtered to have a differential stability value $>= 0.1$. Dots represent individual genes in each pathway. ppp, pentose phosphate pathway; tca, tricarboxylic acid cycle; oxphos, oxidative phosphorylation; lactate, lactate metabolism and transport.
(PDF)

**S1 Table. Energy metabolic pathway gene sets.** Overview of energy metabolic pathways and their final gene sets included in this study. Gene sets were produced based on GO biological processes and Reactome pathway IDs. Genes annotated in both databases used for the analyses are listed. ppp, pentose phosphate pathway; tca, tricarboxylic acid cycle; oxphos, oxidative phosphorylation; Lactate, lactate metabolism and transport.
(PDF)

**S2 Table. Visual ROIs.** Visual regions in the Glasser atlas. Delineations were defined according to the supplementary neuroanatomical results from [59] and https://neuroimaging-core-docs.readthedocs.io/en/latest/pages/atlases.html.
(PDF)

**S3 Table. Cell-type marker genes.** Individual cell-type markers used to produce cortical maps for inhibitory and excitatory neuronal subtypes. Marker genes were obtained from [93,184].
(PDF)

**S4 Table. Age groups for BrainSpan RNA-seq samples.** RNA-sequencing data (Genecode v10 summarized to genes) were downloaded from the BrainSpan database (https://www.brainspan.org/static/download.html/). Samples were binned into eight major developmental stages [93]. Last column shows number of cortical samples for each age group used in the analysis. pcw, post conception weeks; mos, months; yrs, years.
(PDF)

**S5 Table. BrainSpan energy pathway gene sets.** Energy metabolism pathway gene sets used in lifespan trajectory analysis. Note that the same GO biological processes and Reactome pathway IDs were used across all analysis and the difference in final gene sets for each pathway is the result of different gene data availability in AHBA and BrainSpan datasets. PPP, pentose phosphate pathway; TCA, tricarboxylic acid cycle; OXPHOS, oxidative phosphorylation; Lactate, lactate metabolism and transport; Ketone Body, ketone body utilization.
(PDF)

**S6 Table.Energy metabolic pathway gene sets without the differential stability threshold.** As before, gene sets were produced based on GO biological processes and Reactome pathway IDs. Genes annotated in both databases used for the analyses are listed. PPP, pentose phosphate pathway; TCA, tricarboxylic acid cycle; OXPHOS, oxidative phosphorylation; Lactate, lactate metabolism and transport.
(PDF)

**S7 Table. Excluded genes in the OXPHOS map broken down by individual complexes.** Genes not available in the AHBA, as well as genes not fulfilling the differential stability threshold were excluded from the OXPHOS gene expression matrix.
(PDF)

**S8 Table. Age groups for BrainSpan microarray samples.** Microarray data were downloaded from the BrainSpan database (https://www.brainspan.org/static/download.html/) Samples were binned into eight major developmental stages [93]. Last column shows number of cortical samples for each age group used in the analysis. pcw, Post conception weeks; mos, months; yrs, years.
(PDF)

**S9 Table. Gene sets for the extended set of energy pathways.** As before, gene sets were produced based on GO biological processes and Reactome pathway IDs. When available in both databases, consensus genes used to create the energy mean expression maps are listed. PPP, pentose phosphate pathway; TCA, tricarboxylic acid cycle; OXPHOS, oxidative phosphorylation; Lactate, lactate metabolism and transport.
(PDF)

**S1 Data. Supplementary numerical data sheets for Figures 2A-D, 3, S2A,C,D, S4, S9B, S10A-C, and S13.**
(XLSX)

## Acknowledgments

We thank Vincent Bazinet, Eric Ceballos, Asa Farahani, Zhen-Qi Liu, Andrea Luppi, Filip Milisav, Filip Morys and Andrew Vo for their comments and suggestions on the manuscript. We thank all contributors to the open-source datasets used in this study.

## Author contributions

**Conceptualization:** Moohebat Pourmajidian, Bratislav Misic, Alain Dagher.

**Data curation:** Moohebat Pourmajidian, Justine Y Hansen, Golia Shafiei.

**Formal analysis:** Moohebat Pourmajidian.

**Funding acquisition:** Moohebat Pourmajidian, Bratislav Misic, Alain Dagher.

**Investigation:** Moohebat Pourmajidian, Alain Dagher.

**Methodology:** Moohebat Pourmajidian, Justine Y Hansen, Golia Shafiei, Bratislav Misic, Alain Dagher.

**Project administration:** Alain Dagher.

**Resources:** Bratislav Misic, Alain Dagher.

**Software:** Bratislav Misic.

**Supervision:** Bratislav Misic, Alain Dagher.

**Validation:** Moohebat Pourmajidian, Bratislav Misic, Alain Dagher.

**Visualization:** Moohebat Pourmajidian.

**Writing – original draft:** Moohebat Pourmajidian, Bratislav Misic, Alain Dagher.

**Writing – review & editing:** Moohebat Pourmajidian, Justine Y Hansen, Golia Shafiei, Bratislav Misic, Alain Dagher.

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
