## [Editor Report · Decision Letter 0]

11 Jul 2025

Dear Dr Dagher,

Thank you for submitting your manuscript entitled "Mapping energy metabolism pathways in the human brain" for consideration as a Research Article by PLOS Biology. We apologize for the delay as we sought input from an academic editor.

Your manuscript has now been evaluated by the PLOS Biology editorial staff, as well as by an academic editor with relevant expertise, and I am writing to let you know that we would like to send your submission out for external peer review.

Once your full submission is complete, your paper will undergo a series of checks in preparation for peer review. After your manuscript has passed the checks it will be sent out for review. To provide the metadata for your submission, please Login to Editorial Manager (https://www.editorialmanager.com/pbiology) within two working days, i.e. by Jul 13 2025 11:59PM.

Kind regards,

Taylor

Taylor Hart, PhD,

Associate Editor

PLOS Biology

thart@plos.org

---

## [Decision Letter · Decision Letter 1]

2 Sep 2025

Dear Dr Dagher,

Thank you for your patience while your manuscript "Mapping energy metabolism pathways in the human brain" was peer-reviewed at PLOS Biology. It has now been evaluated by the PLOS Biology editors, an Academic Editor with relevant expertise, and by several independent reviewers.

In light of the reviews, which you will find at the end of this email, we would like to invite you to revise the work to thoroughly address the reviewers' reports.

As you will see, the reviewers expressed interest in your approach and found the study valuable. They also suggested further validations and additional analyses, and pointed out areas requiring further explanation or discussion. In your revision, you should carefully consider the points raised by the reviewers and respond to them thoroughly.

Given the extent of revision needed, we cannot make a decision about publication until we have seen the revised manuscript and your response to the reviewers' comments. Your revised manuscript is likely to be sent for further evaluation by all or a subset of the reviewers.

**IMPORTANT - SUBMITTING YOUR REVISION**

*Re-submission Checklist*

*Published Peer Review*

*PLOS Data Policy*

*Blot and Gel Data Policy*

Sincerely,

Taylor

Taylor Hart, PhD,

Associate Editor

PLOS Biology

thart@plos.org

REVIEWS:

Reviewer #1 [Anna Sophia Monzel]: The manuscript by Pourmajidian et al. maps five major energy metabolism pathways (glycolysis, TCA, OxPhos, PPP, and lactate, which they term "energy maps") across human cortical regions at the transcriptomics level, linking them to structural, functional, and developmental features. Using curated gene sets with the Allen Human Brain Atlas microarray data, they relate spatial gene expression patterns to cortical cytoarchitecture, functional connectivity, cell type composition, laminar organization, and PET/MRI metabolic imaging, and extend analyses to developmental trajectories using BrainSpan RNAseq data. This descriptive study, based entirely on publicly available datasets, reveals distinct regional distributions and lifespan changes for each of the five pathways, highlighting differences between ATP-producing and biosynthetic processes, providing a resource with potential relevance for future neurological and neurodegenerative disease research, once validated in larger cohorts. I have no major concerns with the paper, and I find the study valuable.

Strengths:

- Whole-brain profiling of key energy pathways using a pathway-based approach, which is more interpretable than differentially expressed genes alone

- Inclusion of extended mitochondrial-related pathways (ETC complexes, NADH shuttle, ROS…), which I would have liked to see examined and discussed even further (maybe in a follow-up study)

- Linking energy maps to structural, functional, and developmental features

- Overall positive correlation with recently published MitoBrainMap data, especially for ETC complexes and mitochondrial respiratory capacity and their OxPhos score

- Developmental trajectory analyses (though effect sizes would strengthen)

- Thorough and transparent computational methods with clearly documented, well organized code on GitHub

- Provides a valuable macro-scale view and method of oxidative and biosynthetic potential (though transcript-level data cannot establish actual functional capacity).

Minor points:

- Some gene sets are imbalanced (for example, OxPhos is enriched for Complex I genes, only very few CIV genes). It would be useful to see the excluded genes broken down by sub-pathway. The authors addressed this in the end in their sensitivity analysis, where they repeated the analysis with all available genes (which genes were included and found in the dataset?)

- Mitochondrial DNA-encoded transcripts are missing despite their importance for brain energetics. However, those transcripts might not have been available in the AHBA dataset in first place as they are often excluded. Clarification on whether this is due to dataset limitations vs QC exclusions would be helpful.

- I could not find gene lists for additional pathways (FAO, CI, CII,…), they would be useful and also needed for future interpretations

- Some genes are not truly pathway members (eg PRPS2 in the PPP set, which is technically part of nucleotide synthesis)

- There is limited dissection of cell-type specific mitochondrial adaptations, and bulk data may mask fine-grained heterogeneity. The authors addressed this sufficiently in their cell type composition correlation in Figure 5 though.

- The weak OxPhos-CMRO2 correlation is surprising and could be further explored (e.g. testing correlations by individual ETC complex). As the authors point out, transcript abundance does not necessarily reflect functional activity (especially for long-lived mitochondrial proteins), and oxygen consumption may be decoupled from transcription. Also, CMRO2 measurements may underestimate spare respiratory capacity that might be reflected at the mRNA level.

- As I am not a neurologist or brain imaging specialist, I cannot comment on the choice of brain parcellation, functional network definitions, or cytoarchitecture interpretations.

Reviewer #2: This study utilizes transcriptomic data from the Allen Human Brain Atlas (AHBA) alongside manually curated gene sets for metabolic pathways to construct and compare spatial cortical maps of five core energy metabolism pathways: glycolysis, the pentose phosphate pathway (PPP), the tricarboxylic acid (TCA) cycle, oxidative phosphorylation (OXPHOS), and lactate metabolism. These pathway maps are then aligned and analyzed in the context of multiscale structural and functional brain attributes, including network topology, cortical laminar architecture, cell-type specificity, etc. Here are my comments:

1. A recent study (https://www.nature.com/articles/s41586-025-08740-6) provides whole-brain spatial estimates of OXPHOS enzyme activity, mitochondrial DNA/volume density, and mitochondrial respiratory capacity via multimodal MRI-based regression modeling. Please using this dataset to validate and interpret your metabolic pathway maps, especially for OXPHOS-related spatial patterns.

2. Circadian gene expression is strongly influenced by energy constraints (PMID: 37692081). Please describe the spatial distribution and expression profiles of circadian regulators, such as BMAL1, CLOCK, PER, and CRY genes within your cortical pathway maps. Their inclusion could offer mechanistic insight into the interplay between circadian regulation and brain energy metabolism.

3. The final PPP gene set listed in the manuscript ("Energy metabolic pathway gene sets" table) includes only five genes (PGD, PRPS2, RBKS, RPEL1, and RPIA), omitting several well-established key PPP enzymes. Please clarify the rationale for excluding other canonical genes (e.g., G6PD) and discuss how this may impact the completeness or interpretability of the PPP spatial map.

Reviewer #3: Pourmajidian and collaborators present an integrative analysis combining energy metabolism pathways, transcriptomics, and multi-scale structural and functional data of the human brain. Their study provides an interesting example of how different kinds of available data resources can be reused and combined to explore an interesting biological problem: the distribution of energy metabolism pathway potential across the human brain. Among their main findings, they report an heterogeneous spatial distribution of energy metabolism pathways across structural and functional areas as estimated by gene expression, a dichotomy in these patterns between ATP production and anabolic pathways, and (un)expected correlations between pathway gene expression and neurophysiological activity measurements. More broadly, they propose that gene expression can be used to study the metabolic makeup of the human brain across space and developmental time.

The motivation, analyses, and results are presented clearly. I have some comments intended to improve some aspects of the analysis and interpretation.

The authors use multiple neurophysiological datasets to correlate with gene expression. Unlike standard neurophysiological measurements, there is no such thing as a standard cell type composition map. Rather, cell type composition is also inferred from the bulk transcriptomic data they use to estimate metabolic pathway potential. A brief explanation of this contrasting approach and the underlying assumptions would improve the results section presenting this analysis -- which is in between all other standard methods.

Related to the previous point, given the nonstandard measurement, a presentation of cell type estimations across brain regions, how variable and correlated they are, and how these measurements fit or not with expectations based on literature would strengthen the analyses. The authors may consider using similar individual markers to contrast with their estimates as those used in the reference they cite (Wagstyl et al. 66).

The authors find significant positive correlations between inhibitory neurons and all energy metabolism maps. They attribute this association to their fast-spiking activity. However, not all inhibitory neurons are fast-spiking, thus considering only one class likely confounds composition estimates. Based on data on Wagstyl et al. 66, it seems clear that the distribution of markers of specific excitatory and inhibitory neuron subtypes is heterogeneous. This provides an opportunity to more directly contrast to what extent average expression of metabolic pathways correlates with expression of fast-spiking (e.g., PV-expression) neurons and not others. This kind of more targeted analysis would be a nice addition to the study to further support their interpretation/explanation.

The authors analyse the correlation of individual genes within energy pathways and CMRglc and notice the high variability. This suggests that a single average value might not necessarily be a good representation of metabolic potential, but might be biased to specific genes. An additional sensitivity test to address this point would strengthen the analytical choice. The PCA analysis in Fig. S2 relates to this. How consistent are PCs vs average? In a given comparison with a spatial map, how variable are the correlation estimates of individual genes and how much they deviate from the correlation of the average expression. I appreciate that some of these limitations are mentioned in the discussion section.

The authors do not observe a significant correlation between glycolysis/OXPHOS maps and metabolic PET imaging estimates of resting-state glucose and oxygen consumption or cerebral blood flow. They interpret discrepancy as reflecting the fact that "PET measures energy usage at rest while gene expression is a measure of potential energy demand and supply during neuronal activation". Gene expression changes as captured with the resolution of bulk microarray snapshots likely reflect cellular variability and not changes in neuronal activity. Many of the correlations observed are likely explained by the differential expression of enzymes and metabolic pathway genes between the different cell types and subtypes with variable composition across the brain. This observation similarly applies to the developmental changes -- as consistent with the neuronal progenitor vs synapse analysis in Fig. S6. Discussing this additional interpretation in the discussion section would improve the results and interpretation.

Minor technical comment:

The authors may consider including as negative control the expression of genes not expected to vary across brain regions to support the specificity of metabolic pathway changes -- e.g., any housekeeping gene(s).

---

## [Decision Letter · Decision Letter 2]

16 Dec 2025

Dear Dr Dagher,

Thank you for your patience while we considered your revised manuscript "Mapping energy metabolism pathways in the human brain" for publication as a Research Article at PLOS Biology. This revised version of your manuscript has been evaluated by the PLOS Biology editors, the Academic Editor, and the original reviewers.

Based on the reviews, we are likely to accept this manuscript for publication. Please also make sure to address the following data and other policy-related requests.

IMPORTANT: Please ensure that you address the following editorial points:

**Title:

-- We would like to tweak your study's title to emphasize the breadth of analyses. Would you accept the following alternative version?

"Five major metabolic pathways show distinct regional distributions and lifespan changes in the human brain"

**Financial disclosure statement:

-- Please add the grant numbers and links to the funding agencies in the Financial Disclosure statement in the manuscript details.

**Ethics:

My study does not require an ethics statement.

-- We understand that you made use of existing datasets in this study. However, as these data derive from human subjects, please add a brief statement about where the ethical approvals and consent information for the original studies can be obtained.

**Data and Code:

-- Thank you for making the data available through your online repositories. We also request that you provide the numerical data underlying some of the figure panels, which you could provide either as a supplemental item "S1 Data" (upload as S1_Data.xlsx) or within your online deposition. This applies to the following figure panels:

2ABC

3AB

S2CD

S4

S9b(lower right)

S10B(except left)C

S13

-- Please cite the location of the data clearly in all relevant main and supplementary Figure legends, e.g. “The data underlying this Figure can be found in S1 Data” or “The data underlying this Figure can be found in https://doi.org/10.5281/zenodo.XXXXX”

-- We would also prefer if you upload your supplemental information items separately, rather than together in a PDF. This is because supplemental word documents are not proofread and are less likely to be examined by readers. If you opt to keep the supplemental document format, please ensure that all references in the supplement are also included in the main references list, as references that are only found in the supplement will not be counted as citations.

-- Thank you for providing the underlying code in GitHub. However, because Github depositions can be readily changed or deleted, please make a permanent DOI’d copy (e.g. in Zenodo) and provide this URL in the manuscript and Data Availability Statement.

We expect to receive your revised manuscript by January 5.

*Published Peer Review History*

*Press*

Sincerely,

Taylor

Taylor Hart, PhD,

Associate Editor

thart@plos.org

PLOS Biology

REVIEWS

Reviewer #1 [Anna Sophia Monzel]: The authors have sufficiently addressed my suggestions.

Reviewer #2: All my concerns have been addressed.

Reviewer #3: The authors have fully addressed all my concerns. I do not have any further comments.

---

## [Editor Report · Decision Letter 3]

12 Jan 2026

Dear Dr Dagher,

Thank you for the submission of your revised Research Article "Five energy metabolism pathways show distinct regional distributions and lifespan trajectories in the human brain" for publication in PLOS Biology. On behalf of my colleagues and the Academic Editor, Laura Lewis, I am pleased to say that we can in principle accept your manuscript for publication, provided you address any remaining formatting and reporting issues. These will be detailed in an email you should receive within 2-3 business days from our colleagues in the journal operations team; no action is required from you until then. Please note that we will not be able to formally accept your manuscript and schedule it for publication until you have completed any requested changes.

PRESS

Sincerely,

Taylor Hart

Taylor Hart, PhD,

Associate Editor

PLOS Biology

thart@plos.org